# Consistent Spectral Clustering under Hyperbolic Geometry

## Abstract

Spectral clustering is a widely used unsupervised learning method that partitions data by analyzing the spectrum of a similarity graph, where the classical formulations implicitly assume Euclidean geometry. But this assumption becomes inadequate when data exhibit a hierarchical or tree-like structure. In such settings, Euclidean distances distort geodesic relationships, leading to unstable spectral embeddings and degraded clustering performance. Motivated by this limitation, we study spectral clustering under hyperbolic geometry, a natural model for hierarchical data, and propose an intrinsically hyperbolic spectral clustering framework in which the similarity operator is defined using hyperbolic distances after estimating a latent hierarchical root. This construction yields a hyperbolic graph Laplacian whose spectrum better reflects the underlying geometry of the data. We provide a rigorous theoretical analysis establishing the weak consistency of the proposed method under a hyperbolic latent variable model, with convergence rates at least as fast as the classical spectral clustering in Euclidean space. Empirical results on real-world hierarchical datasets demonstrate competitive performance relative to other existing deep and hierarchical clustering benchmarks, highlighting the importance of geometric modeling in spectral methods, and positioning hyperbolic geometry as a principled foundation for clustering complex structured data.

## 1 Introduction

In the realm of machine learning, the pivotal process of categorizing data points into cohesive groups remains vital for uncovering patterns, extracting insights, and facilitating various applications, ranging from customer segmentation to anomaly detection and image understanding Ezugwu et al. (2022). Among the paradigms of clustering algorithms Filippone et al. (2008), alongside *Partitional, Hierarchical,* and *Density-based* techniques, *Spectral Clustering* on Euclidean spaces has garnered extensive research attention Von Luxburg (2007). Spectral clustering operates in the spectral domain, utilizing the eigenvalues and eigenvectors of the Laplacian of a similarity graph constructed from the data. This algorithm initially constructs a similarity graph, where nodes represent data points and edges indicate pairwise similarities or affinities among the data points. It then computes the graph Laplacian matrix, capturing the graph's structural properties and encoding relationships among the data points. Since its inception [See Donath & Hoffman (1973) and Fiedler (1973)], the Euclidean version of Spectral Clustering has significantly evolved. In the simplest two-cluster setting, spectral clustering uses the eigenvector corresponding to the second smallest eigenvalue of the graph Laplacian (the Fiedler vector), and assigns cluster labels based on the sign of its entries. More generally, one may consider the first $k$ eigenvectors, treat the rows of the resulting matrix as embedded points, and apply $k$-means clustering; however, for $k = 2$, this reduces essentially to thresholding the second eigenvector. This particular form of spectral clustering finds applications Suryanarayana et al. (2015) in Speech Separation Bach & Jordan (2006), Image Segmentation Tung et al. (2010), Text Mining Dhillon (2001), VLSI design Hagen & Kahng (1992), and more. A comprehensive tutorial is also available at Von Luxburg (2007). Here, we will briefly review the most commonly used Euclidean Spectral Clustering Algorithm.

When connectedness is a crucial criterion for clustering algorithms, the conventional form of Spectral Clustering emerges as a highly effective approach. It transforms the standard data clustering problem in a given

Euclidean space into a graph partitioning problem by representing each data point as a node in the graph. Subsequently, it determines the dataset labels by discerning the spectrum of the graph. Beginning with a set of data points $X := \{x_1, x_2, ..., x_n\} \in \mathbb{R}^d$. A symmetric similarity function (also known as the kernel function) $k_{i,j} := k(x_i, x_j)$, along with the number of clusters $p$, we construct the similarity matrix $W := w(i, j) = k_{ij}$. In its simplest form, Spectral Clustering treats the similarity matrix $W$ as an adjacency matrix of a latent graph like structures, and aims to partition the graph to minimize the sum of weights across the edges of the two partitions. Mathematically, we try to solve an optimization problem Von Luxburg (2007) of the following form:

$$\min_{U \in \mathbb{R}^{n \times p}} C := \min_{U \in \mathbb{R}^{n \times p}} Tr(U^\top L'U) \text{ s.t. } U^\top U = I_p, \tag{1}$$

where $L := D - W$ is the Graph Laplacian and the degree matrix $D := diag(d_1, d_2, , , , d_n), d_i := \sum_{j=1}^n w(i, j)$ and $L' := D^{-1/2}LD^{-1/2} = I - D^{-1/2}WD^{-1/2}$. $U$ is a label feature matrix, $p$ being the number of clusters. Then, the simplest form of spectral clustering aims to minimize the trace by the feature matrix $U$ by considering the first $p$ eigenvectors of $L'$ as its rows. Finally, we return the labels to the original data points in the order they were taken to construct the degree matrix $W$. Some of the variants of this algorithm can be found at Von Luxburg (2007).

Despite its success, Euclidean spectral clustering implicitly assumes that the underlying data geometry is well approximated by Euclidean space. This assumption becomes increasingly restrictive when data exhibit hierarchical, tree-like, or graph-structured organization. In such cases, Euclidean distances distort geodesic relationships Nadler & Galun (2006), flatten hierarchical depth Tasdemir et al. (2014), and collapse spectral gaps Yu et al. (2019), leading to unstable embeddings and degraded clustering performance. These limitations are particularly pronounced in modern datasets arising from networks, ontologies, and biological systems, where hierarchy is intrinsic rather than incidental.

Hyperbolic geometry provides a natural alternative for modeling hierarchical data, as it allows exponential volume growth and can represent tree-like structures with low distortion even in low dimensions. Recent work has demonstrated the effectiveness of hyperbolic representations in deep learning models, particularly in computer vision and graph representation learning Peng et al. (2021a); Ganea et al. (2018); Chen et al. (2022); Chami et al. (2019). However, clustering algorithms and non-deep learning methods that operate intrinsically in non-Euclidean spaces remain relatively under-explored. In particular, the interaction between spectral methods and hyperbolic geometry has received limited theoretical and algorithmic attention.

In this work, we study spectral clustering under hyperbolic geometry and propose an intrinsically hyperbolic spectral clustering framework designed to respect hierarchical structure. Our approach replaces the Euclidean similarity matrix with a hyperbolic similarity operator constructed using hyperbolic distances after translating data points with respect to an estimated latent root. This leads to a hyperbolic graph Laplacian whose spectral properties are better aligned with the underlying geometry of the data. Beyond algorithmic design, we provide a theoretical analysis establishing weak consistency of the proposed method under a hyperbolic latent variable model, with convergence rates comparable to classical spectral clustering in Euclidean space. Empirical results on real-world hierarchical datasets demonstrate robust performance across varying hierarchy depths relative to existing deep and non-deep hierarchical clustering algorithms, highlighting the value of geometric modeling in spectral methods.

**Contributions.** Our main contributions are as follows:

- We propose a scalable spectral clustering algorithm on hyperbolic spaces in which an appropriate hyperbolic similarity matrix replaces the Euclidean similarity matrix, after suitably translating the points with respect to an estimated root node of the hierarchy.

- We also provide a theoretical analysis concerning the weak consistency of the algorithm and prove that it converges (in the sense of distribution) at least as fast as the spectral clustering on Euclidean spaces.

- We present simulations pertaining to several real-world hierarchical datasets of our algorithm and compare the results with some of the modern deep and hierarchical clustering algorithms.

Having said that, we organize the rest of our paper in the following way. In Section 2, we will briefly overview the works related to Euclidean Spectral Clustering and its variants. We will also discuss why we need to consider a hyperbolic version of Euclidean Spectral Clustering. Section 3 lays out the mathematical backgrounds of our proposed algorithm. We will discuss several results which will enable us to formulate the algorithm rigorously. We give the details of our proposed algorithm in Section 4. We discuss the motivation behind the steps related to our algorithm. Section 5 has been dedicated to proving the weak consistency of the proposed algorithm. Section 6 presents and discusses the experimental results. We discuss the limitations and a possible extension to our SHSC algorithm towards multi-rooted formulation in Section 7. Finally, conclusions are drawn in Section 8.

## 2 Related Works

We commence with a brief overview of prominent variants of Euclidean spectral clustering:

1. **Bipartite Spectral Clustering on Graphs (ESCG)**: Introduced by Liu et al. Liu et al. (2013) in 2013, this algorithm primarily aims to reduce the time complexity during spectral decomposition of the affinity matrix by appropriately transforming the input similarity matrix of a graph dataset. The method involves randomly selecting $d(\ll n)$ seeds from a given graph of input size $n$, followed by generating $d$ supernodes using Dijkstra's Algorithm to find the shortest distance from the graph nodes to the seeds. This process reduces the size of the similarity matrix $\tilde{W} := RW$, where $R$ is the indicator matrix of size $d \times n$ and $W$ is the original affinity matrix. Subsequently, it proceeds with spectral decomposition of the normalized $Z := D_2^{-1/2}\tilde{W}D_1^{-1/2}$, where $D_1$ and $D_2$ are diagonal matrices containing the column and row sums of $\tilde{W}$, respectively. The algorithm computes the $k$ largest eigenvectors of $ZZ^t$ and generates $k$ clusters based on the $k$-means algorithm on the matrix $U := D_1^{-1/2}X$, where $X$ is the right singular matrix in the singular value decomposition of $Z$.

2. **Fast Spectral Clustering with approximate eigenvectors (FastESC)**: Developed by He et al. He et al. (2018) in 2019, this algorithm initially performs $k$-means clustering on the dataset with several clusters greater than the true clusters and then conducts spectral clustering on the centroids obtained from the $k$-means. Similar to ESCG, this algorithm also focuses on reducing the size of the input similarity matrix for spectral clustering.

3. **Low Rank Representation Clustering (LRR)**: Assuming a lower-rank representation of the dataset $X := [x_1, x_2, ..., x_n]$, where each $x_i$ is the $i$-th data vector in $\mathbb{R}^D$, this algorithm aims to solve an optimization problem to minimize the rank of a matrix $Z$ subject to $X = AZ$. Here, $A = [a_1, a_2, ..., a_m]$ is a dictionary, and $Z := [z_1, z_2, ..., z_n]$ is the coefficient matrix representing $x_i$ in a lower-dimensional subspace. The algorithm iteratively updates $Z$ and an error matrix $E$ as proposed by Liu et al. in Liu et al. (2010).

4. **Ultra-Scalable Spectral Clustering (U-SPEC)**: Among other variants of Spectral Clustering, such as Ultra-Scalable Spectral Clustering Algorithm (U-SPEC) Huang et al. (2019) or Constrained Laplacian Rank Clustering (CLR) Nie et al. (2016), the primary objective remains consistent - to enhance efficiency by reducing the burden of the spectral decomposition step through minimizing the size of the input similarity matrix. However, there has been minimal exploration regarding the translation of these algorithms into a hyperbolic setup. In this context, this marks the initial attempt to elevate non-deep machine learning algorithms beyond Euclidean Spaces.

Datasets with hierarchical structure are naturally represented as trees, where nodes are organized from a root toward increasing depth. As we move away from the root, the distance between nodes at the same depth but belonging to different branches grows exponentially with respect to their height. This exponential expansion makes Euclidean space ill-suited for representing hierarchical data, as Euclidean distances cannot faithfully preserve such growth without incurring severe distortion. In contrast, hyperbolic spaces exhibit exponential volume growth with distance from the origin, making them a natural geometric model for hierarchical and tree-like structures. This observation has motivated a growing body of work on hyperbolic representations,

particularly within the context of deep neural networks, where hyperbolic embeddings have been shown to improve representation learning for structured data Peng et al. (2021b); Ganea et al. (2018); Chami et al. (2019). More recently, in the context of downstream self-supervised learning, Long & van Noord (2023) proposed scalable Hyperbolic Hierarchical Clustering (sHHC), which learns continuous hierarchies in hyperbolic space and constructs hierarchical pseudo-labels from audio and visual data, achieving competitive performance in activity recognition. Despite these advances, clustering methods that operate intrinsically in hyperbolic space and are not tied to deep learning architectures remain relatively under-explored. In this work, we address this gap by proposing a general-purpose spectral clustering framework defined on a chosen hyperbolic space, obtained by embedding the original Euclidean data in a manner that preserves the underlying hierarchical structure with minimal distortion.

## 3 Preliminaries

We will briefly explore the fundamentals of Riemannian Geometry and Gromov Hyperbolicity, which form the basis of our proposed algorithm.

**Riemannian Manifold:** Mathematically, a *Manifold* $\mathcal{M}$ of dimension $n$ is a topological space which is second countable, Hausdorff, and is locally homeomorphic to a subset of $\mathbb{R}^n$ Tu (2017). For each $a \in \mathcal{M}$, the *Tangent Space* $T_a(\mathcal{M})$ can be thought of as an attached linear differentiable manifold having the same dimension of the ambient space [a manifold along with a differentiable structure] with an additional vector space structure, more specifically, as a linear approximation of $\mathcal{M}$ at $a$. $\mathcal{M}$ is termed as a *Riemannian Manifold* if for every point $a \in \mathcal{M}$, there exists a collection of smoothly varying metric tensors $g := g_a : T_a(\mathcal{M}) \times T_a(\mathcal{M}) \to \mathbb{R}, a \in \mathcal{M}$ do Carmo (1992). The distance function on this space is induced by these collections of metrics, which is a function between two points $p, q \in \mathcal{M}$ joined by a piecewise smooth curve $\gamma : [0,1] \to \mathcal{M}$ with $\gamma(0) = p$ and $\gamma(1) = q$, where the distance from $p$ to $q$ is calculated as $L(\gamma) := \int_0^1 g_{\gamma(t)}(\gamma'(t), \gamma'(t))^{1/2} dt$. We consider the set of all curves between two points and will consider the curve for which the distance between them is the minimum and call that curve as a *Geodesic* and its length is designated as the *Geodesic Distance* [Geodesic between two points may not be unique, but the Geodesic Distance is]. For two linearly independent vectors $u$ and $v$ at $T_a(\mathcal{M})$, the *Sectional Curvature* at $a$ is defined as $k_a(u,v) := \frac{g_a(R(u,v)v,u)}{g_a(u,u)g_a(v,v) - g_a(u,v)^2}$, where $R$ is the Riemannian Curvature Tensor or the Riemannian Connection defined as $R(u,v)w := \nabla_u \nabla_v w - \nabla_v \nabla_u w - \nabla_{(\nabla_u v - \nabla_v u)} w$, with $\nabla_u v$ being the directional derivative of $v$ in the direction of $u$.

**Hyperbolic Space:** Following these notations, *Hyperbolic Space* of dimension $n$ is defined as a complete and connected Riemannian Manifold with constant negative sectional curvature. Various theoretical models of Hyperbolic Spaces have been proposed, such as the Poincaré Half-Space Model, Poincaré ball Model, Hyperboloid Model [also known as the Minkowski Model], and Klein-Beltrami Model. Nevertheless, the renowned *Killing-Hopf Theorem* Lang (1995) asserts that all model hyperbolic spaces are isometric, given they share the same dimension and curvature. We will leverage this theorem to develop our proposed algorithm uniquely within a specific model space, thereby avoiding performance variations. For convenience, we select the Poincaré ball Model.

**Poincaré Ball Model:** One can visualize the Poincaré ball or Poincaré Ball of dimension $n$ with curvature $k(<0)[c=-k]$ as a ball of radius $1/\sqrt{c}$ embedded in $\mathbb{R}^n$ Lee (2006). The geodesics in this model are circular arcs that intersect orthogonally with the spherical surface of this ball. The geodesic distance between $a$ and $b$ (where $\|a\|, \|b\| < 1/\sqrt{c}$) is given by

$$d(a,b) := 2\sinh^{-1}\left(\sqrt{2\frac{\|a-b\|^2}{c(\frac{1}{c} - \|a\|^2)(\frac{1}{c} - \|b\|^2)}}\right). \tag{2}$$

Throughout the rest of our paper, $\mathbb{D}_c^n$ will indicate the $n$-dimensional Poincaré Ball with curvature $-c[c>0]$.

**Gyrovector Space:** The idea of a Gyrovector space, put forth by Ungar [see Ungar (2022)], provides a framework for examining the vector space structures within Hyperbolic Space. This concept enables the definition of unique addition and scalar multiplication operations rooted in weakly associative gyrogroups. For an in-depth exploration, we refer to Vermeer's work Vermeer (2005).

In this setup, it is essential to talk about Möbius Gyrovector Addition and Möbius Scalar Multiplication on the Poincaré ball. The inherent isometric transformations between hyperbolic spaces of the same dimension allow these multiplicative and additive structures to be applied to other model hyperbolic spaces (refer to Ungar (2022)). These operations will be essential for computing the Fréchet Centroid in Algorithm 1.

1. **Möbius Addition:** We define the Möbius addition of two points $p$ and $q$ on the Poincaré ball as:

$$p \oplus_c q := \frac{(1 + 2c\langle p, q\rangle + c\|q\|^2)p + (1 - c\|p\|^2)q}{1 + 2c\langle p, q\rangle + c^2\|p\|^2\|q\|^2}, \tag{3}$$

where $c$ is the negative of the curvature of the Poincaré ball.

2. **Möbius Scalar Multiplication:** We also define the scalar multiplication of a $r \in \mathbb{R}$, $c > 0$ and $p$ in the Poincaré ball as:

$$r \otimes_c p := \frac{1}{\sqrt{c}} \tanh\left(r \tanh^{-1}(\sqrt{c}\|p\|)\right) \frac{p}{\|p\|}. \tag{4}$$

This addition and scalar multiplication satisfy the Gyrovector Group Axioms [see Ungar (2022)].

**Fréchet Centroid:** For a set of $m$ points $\{x_1, x_2, ..., x_m\} \in \mathbb{D}_c^n$, we define the Fréchet centroid as a generalized notion of the Euclidean Centroid, defined as

$$FC(x_1, x_2, ..., x_m) := \frac{1}{m} \otimes_c (x_1 \oplus_c (x_2 \oplus_c ... (x_{m-1} \oplus_c x_m))). \tag{5}$$

**Exponential & Logarithmic Maps:** For a point $p \in \mathbb{D}_c^n$, the Exponential Map $\exp_p^c : T_p(\mathbb{D}_c^n) \subseteq \mathbb{R}^n \to \mathbb{D}_c^n$ projects a point from the tangent space of the Poincaré ball to the Poincaré ball itself along the direction of the unit speed geodesic starting from $p \in \mathbb{D}_c^n$ in the direction of $v \in T_p(\mathbb{D}_c^n)$. While the Logarithmic Map, $\log_p^c : \mathbb{D}_c^n \to T_p(\mathbb{D}_c^n) \subseteq \mathbb{R}^n$ performs the inverse operation by projecting a point back to the tangent space at $p \in \mathbb{D}_c^n$ from the Poincaré ball, along the reverse geodesic outlined by the Exponential Map. Their mathematical expressions are given as follows:

$$\exp_p^c(q) := p \oplus_c \left(\tanh\left(\sqrt{c}\frac{\lambda_p^c\|q\|}{2}\right) \frac{q}{\sqrt{c}\|q\|}\right) \tag{6}$$

and

$$\log_p^c(z) := \frac{2}{\sqrt{c}\lambda_p^c} \tanh^{-1}\left(\sqrt{c}\| - p \oplus_c z\|\right) \frac{-p \oplus_c z}{\| - p \oplus_c z\|}, \tag{7}$$

for $z \neq p$ and $q \neq 0$ and the Poincaré conformal factor $\lambda_p^c := \frac{2}{(1-c\|p\|^2)}$.

**Gromov Hyperbolicity:** For any metric space $(X, d)$, one defines the *Gromov Product* of two points $a, b$ with respect to a third point $c$ as

$$(a, b)_w := \frac{1}{2}(d(a, c) + d(b, c) - d(a, b))$$

and we say $X$ is $\delta-$ hyperbolic iff for any tuple $(a, b, c, w)$ of four points in $X$, we have

$$(a, c)_w \geq \min((a, b)_w, (b, c)_w) - \delta. \tag{8}$$

It can be proved that if Equation 8 is satisfied for one base point $w$, then it is satisfied for all base points up to a constant multiple of 2 Coornaert et al. (2006). Therefore, we can conveniently remove the base point from the definition of the Gromov Product. Although this form was originally introduced by Gromov himself, there

is an equivalent definition of the same easier for implementation purposes, which was introduced by Rips Bridson & Haefliger (2013). It reduces the four-point definition to an arbitrary geodesic triangle $[x, y, z] \in X$. According to Rips Bauer & Roll (2021), such a triangle is said to be $\delta-$ slim if $\delta$ is the minimum positive value such that any side can be contained in the union of $\delta$ neighborhoods of the other two sides, and $X$ is $\delta-$ hyperbolic if any triangle in $X$ is $\delta-$ slim. Rips Bauer & Roll (2021) also showed that there exists a constant $a$ such that Bridson & Haefliger (2013) $X$ is $\delta-$ hyperbolic as defined by Gromov if and only if $X$ is $a \cdot \delta$ hyperbolic as defined by Rips, and we call this $\delta$ as the *Gromov Hyperbolicity Index (GHI)* of $X$. However, according to both definitions, a space with a lower GHI will be more hyperbolic (for a lower GHI, the sides of any geodesic triangle will be closer to each other, indicating more negative bendness/curvature) compared to a space with higher GHI.

**Gromov Hyperbolicity and Its Relation to Tree-Based Hierarchy.** A fundamental challenge in embedding tree-structured hierarchies into Euclidean space is that Euclidean geometry fails to capture the exponential growth of distances induced by tree depth. In a rooted tree, the distance between nodes increases exponentially with depth from the root, whereas Euclidean distances grow only linearly, resulting in significant distortion when representing hierarchical structure. Hyperbolic spaces, by contrast, naturally accommodate exponential distance growth with respect to the distance from the origin, making them well suited for embedding tree-like data.

When a tree is embedded into a hyperbolic space, its geometry is not uniformly hyperbolic across all regions. In particular, the effective Gromov hyperbolicity is smallest in neighborhoods close to the root and increases as one moves farther away. Intuitively, the local neighborhood of the root exhibits the strongest hyperbolic behavior, since paths to different branches share a common trajectory for a longer portion before diverging. In contrast, nodes located deeper in the hierarchy diverge earlier, resulting in larger Gromov hyperbolicity constants.

More precisely, the Gromov hyperbolicity of two nodes (other than the root) with respect to the root quantifies how long the corresponding geodesic paths remain close before separating. Nodes that share a significant portion of their path from the root exhibit smaller hyperbolicity, whereas nodes belonging to distinct subtrees diverge rapidly and exhibit larger hyperbolicity. Consequently, triplets of points with small Gromov hyperbolicity are most informative for identifying the location of the root in a hierarchical structure. Among all triplets drawn from a dataset $X = \{x_1, x_2, \ldots, x_n\}$, those with lower Gromov hyperbolicity contribute more strongly to accurate root estimation. An illustrative example is provided in Figure 1.

**CAT(0) Space.** A geodesic metric space $(\mathcal{X}, d)$ is called a CAT(0) Space if for any $p, q, r \in \mathcal{X}$ and for any length minimizing geodesic $\gamma : [0, 1] \to \mathcal{X}$ with $\gamma(0) = p$ and $\gamma(1) = q$,

$$d^2(r, \gamma(t)) \leq (1 - t)d^2(r, p) + td^2(r, q) - (1 - t)td^2(p, q)$$

holds for all $t \in (0, 1)$. Any complete Riemannian Manifold with non-positive sectional curvature is a CAT(0) Space [see Bacák (2014)]. Therefore, any Hyperbolic Spaces, in particular, the Poincaré Balls, are also CAT(0) Spaces.

Having all the required preliminaries, we are now in a position to describe our proposed algorithm.

## 4 Proposed Method

### 4.1 Motivating Example

Hierarchical datasets, such as those arising in phylogenetic analysis, taxonomies, or nested social networks, present a unique challenge for conventional clustering methods. In these datasets, data points are organized in a tree-like structure where distances between nodes grow exponentially with the depth of the hierarchy. Standard Euclidean clustering methods, including spectral clustering, often fail to capture such nested relationships because Euclidean distances do not scale naturally with the exponential separation of hierarchical levels.

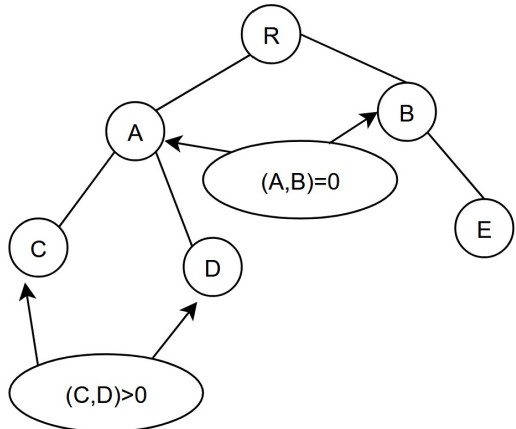

Figure 1: In this tree based hierarchy, the GHI between $A$ and $B$ with respect to $R$ is 0, since they diverge from the root itself. On the other hand, the same between $(C, D)_R$ is more than 0, because they have a common ancestor $A$, which is not the root. Therefore, the positions of $A$ and $B$ will have a higher contribution in determining the position of the root $R$. Similarly, the pairs $(C, E)$ or $(D, E)$ will also contribute highly to determining the location of $R$. Any path between two nodes with no common ancestor other than the root will have the root node on it. Therefore, the lower the GHI between two nodes, the higher the contribution will be in determining the root.

To illustrate the empirical behavior of our SHSC method (**Algorithm 1** and **Algorithm 2**), we consider the HTRU2 dataset from the UCI Machine Learning Repository. This dataset consists of $17,898$ pulsar candidates, each represented by 8 continuous features, and is formulated as a binary classification problem with two clusters: a minority class of real pulsars and a majority class of spurious signals.

We emphasize that this dataset is not known to possess an explicit hierarchical or tree-structured organization. Instead, our use of hyperbolic methods is motivated by an empirical geometric observation: under the chosen Euclidean metric, the dataset exhibits a relatively low Gromov hyperbolicity (GHI $\approx 0.017$), suggesting that its pairwise distance structure can reasonably be well approximated by a tree like geometry. This observation does not imply an intrinsic hierarchical model, but rather indicates that hyperbolic representations may provide a useful inductive bias.

Consistent with this perspective, we compare clustering performance in Euclidean and hyperbolic settings. Euclidean spectral clustering achieves a mean NMI of 0.142, whereas SHSC yields a substantially higher NMI of 0.371 [The normalized mutual information (NMI) Estévez et al. (2009) measures the agreement between two clusterings by comparing the mutual information between their label assignments relative to their individual entropies. For $U$ and $V$ denoting two clusterings, and let $P(i)$, $P(j)$, and $P(i, j)$ be the marginal and joint probabilities defined from the contingency table, i.e.,

$$P(i) = \frac{a_i}{n}, \quad P(j) = \frac{b_j}{n}, \quad P(i, j) = \frac{n_{ij}}{n},$$

Here, $n_{ij}$ is the number of data points shared between cluster $i$ in $U$ and cluster $j$ in $V$, $a_i = \sum_j n_{ij}$, $b_j = \sum_i n_{ij}$, and $n$ is the total number of data points. The entropy of a clustering $U$ is defined as

$$H(U) = -\sum_i P(i) \log P(i),$$

and similarly for $V$,

$$H(V) = -\sum_j P(j) \log P(j).$$

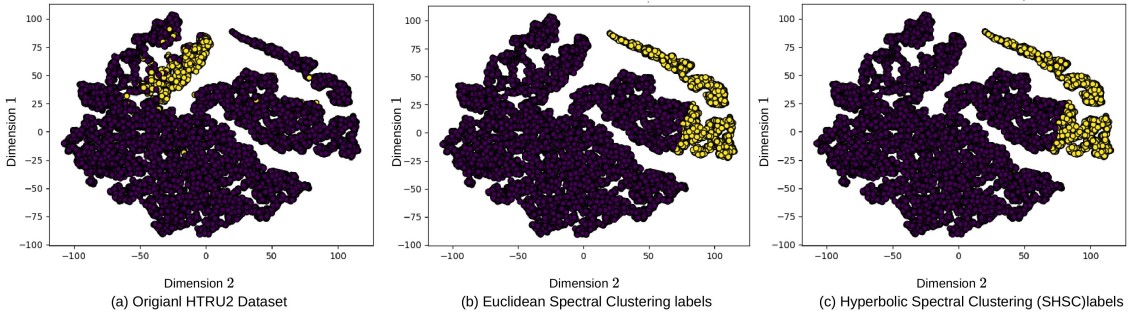

(a) Origianl HTRU2 Dataset  (b) Euclidean Spectral Clustering labels  (c) Hyperbolic Spectral Clustering (SHSC)labels

Figure 2: From Left to Right: tSNE Visualization of the (a) HTRU Dataset, (b) Spectral Clusters, and (c) SHSC Clusters

The mutual information between $U$ and $V$ is given by

$$I(U;V) = \sum_{i,j} P(i,j) \log \frac{P(i,j)}{P(i)\,P(j)}.$$

These results suggest that, even in the absence of an explicit hierarchy, exploiting approximate tree-like geometry can lead to improved clustering performance. We refer to Figure 2 for a 2− dimensional t-SNE visualization of the clusterings resulted by the conventional Euclidean Spectral Clustering (b) and the clustering labels produced by our SHSC (c) algorithm. Evidently, our SHSC performs better in capturing the minority class of real pulsar signals.

Now, let us examine a dataset where the assumptions of hyperbolic or tree-like structure may not hold. We consider the well-known **IRIS dataset**, which is publicly available at the UCI Machine Learning Repository. We first computed the **Gromov Hyperbolicity Index (GHI)** of this dataset using the standard Euclidean metric and obtained a value of approximately 0.695. Such a relatively high GHI suggests that the IRIS dataset does not naturally conform to a hyperbolic or tree-like geometric structure.

To further explore the implications of this observation, we evaluated the clustering performance on this dataset using both our **Scalable Hyperbolic Spectral Clustering (SHSC)** method and the conventional **Euclidean Spectral Clustering**. The results show a noticeable difference: the mean Normalized Mutual Information (NMI) of the cluster labels obtained via Euclidean spectral clustering is approximately 0.8, indicating a strong agreement with the ground-truth labels. In contrast, the mean NMI achieved by our SHSC algorithm is only around 0.47, reflecting a reduced clustering quality in this context.

A visual comparison of the clustering results is presented in Figure 3, highlighting how the SHSC clusters deviate more significantly from the true classes when the dataset lacks an inherent hyperbolic or tree-like structure. These findings illustrate that a higher GHI can correlate with poorer clustering performance for hyperbolic embedding-based methods such as SHSC, emphasizing the importance of dataset geometry in determining the suitability of hyperbolic clustering approaches.

These two examples clearly demonstrates that hyperbolic geometry is well suited for hierarchical data [even for datasets without any inherent hierarchical structure, still whose GHI is low], and motivates the development of SHSC as a scalable and effective method for clustering in such non-Euclidean spaces.

Motivated by the preceding example, we next describe our proposed Scalable Hyperbolic Spectral Clustering (SHSC) framework.

## Scalable Hyperbolic Spectral Clustering (SHSC)

The main problem of working with a hierarchical dataset $\mathcal{X} := \{x_1, x_2, ..., x_n\} \in \mathbb{R}^d$ is to estimate the position of the root node and then embedding the entire dataset with respect to the root in the form of a tree-based hierarchy . To break this bottleneck, we begin with the entire dataset $\mathcal{X}$ being embedded into $\mathbb{D}_c^d$

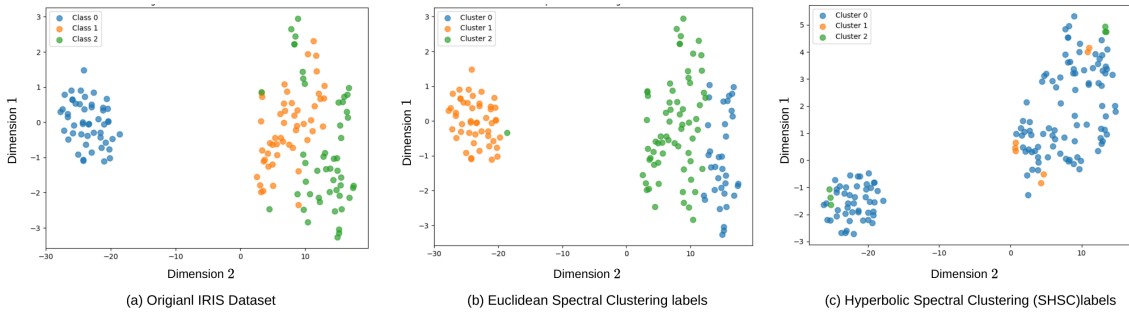

Figure 3: From Left to Right: tSNE Visualization of the (a) IRIS Dataset, (b) Spectral Clusters, and (c) SHSC Clusters

via the $\exp_0^c$ map (i.e. for each $x_i \in \mathcal{X}$, we will consider their embedding $\exp_0^c(x_i) \in \mathbb{D}_c^d$) and then propose a method namely, *Root Estimation using the lowest K Gromov Hyperbolicity Indices* to estimate the position of the root node as per our discussions in Section 3. However, implementing this method is hindered by its inherent complexity of $\mathcal{O}(n^3)$, which comes from considering all 3 combinations of $n$ points, making it difficult to employ for large datasets. We can deal with this problem by introducing a bootstrap method of repeatedly selecting a smaller sample from all datasets and using the lowest $K$ Gromov Hyperbolicity Indices each time. Then, we will translate each data point with respect to this root node once we have an estimate of its position. Following that we will use a scalable hyperbolic spectral clustering and finally assign cluster labels to all data points on a nearest-neighbor basis. The complexity of the entire algorithm will be much less than $\mathcal{O}(n^3)$ as we will eventually see, which is much more efficient compared to the original spectral clustering algorithm. Therefore, we decompose our proposed algorithm into three separate parts:

1. Root Estimation using lowest $K$ Gromov Hyperbolicity Indices,

2. Performing a Scalable Spectral Clustering Algorithm and

3. Final Cluster label Assignment on a Nearest Neighbor Basis.

### 4.1.1   Root Estimation using the lowest K Gromov Hyperbolicity Indices

As discussed in the previous paragraph, we begin with a set of data points $\mathcal{X} \coloneqq \{x_1, x_2, ..., x_n\} \in \mathbb{D}_c^d$. Now we will estimate the position of the root using our comments in 3. As mentioned, we can use those 3 points for which the GHI will be the lowest in $\mathcal{X}$. But to make it accurate, we will consider the $K-$ sets, each consisting of 3 points such that they have the lowest $K-$ GHIs. Finally, we will compute their Fréchet Centroid 5 and accept that as our estimation of the root.

**An Efficient Bootstrap Method for Estimating Root** For a dataset $\mathcal{X}$ with $n$ points, it will require a complexity of $\mathcal{O}(n^3)$ to compute the GHIs for all 3 combinations of the $n$ points. To determine the lowest $K$ many GHIs and the corresponding sets of 3 points requires another complexity of $\mathcal{O}(\log(n))$ (for sorting those GHIs). Therefore, invoking the greedy strategy to find the data points corresponding to the lowest $K$ GHIs can make the algorithm infeasible for large datasets. To mitigate this issue, here we propose a bootstrapping method for estimating the root. From the set of $n$ points, we will randomly select $p'$ many points, where $p < p' \ll n$ ($p$ being the number of clusters). For those $p'$ points, we will find the $K$ sets of 3 points corresponding to the lowest GHIs. We can repeat the entire sampling procedure for $t$ many iterations. To this end, we have a total of $3Kt$ many points, whose Fréchet Centroid 5 will be our estimate for the root [While the Fréchet mean may coincide with a data point in certain cases, this has no impact on our results, since our analysis depends only on its stability characterization.] The entire procedure has a complexity of $\mathcal{O}((p')^3 t + t \log p') = \mathcal{O}((p')^3 t) << \mathcal{O}(n^3)$. We present the pseudocode in **Algorithm 1**.

---

**Algorithm 1** Root Estimation using lowest $K$ Gromov Hyperbolicity Indices

---

**Input:** Dataset $\mathcal{X} := \{x_1, x_2, ..., x_n\} \in \mathbb{R}^d$

**Hyperparameters:** Bootstrap Sample Size $= p'$, Bootstrap Iterations $= t$, Number of lowest GHIs $= K$, Curvature $= -c(c > 0)$ of the Poincaré ball.

**Output:** An estimate of the root node.

1: Obtain the transformed set of points $\mathcal{X}' := \{x_1', x_2', ..., x_n'\} \in \mathbb{D}_c^d$ such that $x_i' := \exp_0^c(x_i)$.
2: **for** iterations$= 1, 2, ..., t$ **do**
3:   pick a bootstrap sample of size $p'$, i.e. a random subset $\mathcal{Y} \subseteq \mathcal{X}'$ of size $p'$.
4:   for each 3 combinations of points of $\mathcal{Y}$, calculate the GHIs.
5:   select the lowest $K$ GHIs and their corresponding 3 points for each of those GHIs.
6: **end for**
7: Compute the Fréchet Centroid of $3Kt$ many points by 5 as an estimate of the root.

---

### 4.1.2 Performing a Scalable Spectral Clustering Algorithm

Once we have the estimate for the root, let's call it $r \in \mathbb{D}_c^d$, we will re-embed each point again via the Möbius Addition, i.e., our transformed points will now be $x_i' := -r \oplus_c x_i \in \mathbb{D}_c^d$ [3]. Now we will perform a scalable spectral clustering on the set of transformed points $\mathcal{X}' := \{x_1', x_2', ...x_n'\}$. But as we know from the conventional spectral clustering algorithm, performing the eigendecomposition on the $n \times n$ normalized Laplacian matrix has a complexity of $\mathcal{O}(n^3)$, which makes it extremely difficult for the implementation on large datasets. Here, we propose a scalable method for performing spectral clustering similar to Huang et al. (2019), but we will not take any hybrid representation at first, and then a $k-$ means clustering on the set of hybrid representatives, followed by a spectral decomposition on a smaller dimensional matrix. We will randomly select a set of $f$ points from the set of $n$ points (for better result, we recommend $f \approx \mathcal{O}(\log n)$, for example, $f = [10 \log n]$ or $f = [20 \log n]$). We will construct the $f \times f$ similarity matrix by considering their pairwise Poincaré Distances and will perform the spectral decomposition of it and cluster those $f$ representatives.

### 4.1.3 Final Cluster Label Assignment on a Nearest Neighbour Basis

Finally, we will assign cluster labels to the rest of the points based on the nearest neighbor. For each point $x_i'$, we will assign its label as the same label of its nearest point among those representatives of $f$ points, i.e. if $f$ is the nearest neighbor of $x_i'$ and the label of $f$ is $j$ for $j \in \{1, 2, ..., p\}$, then we assign the label $j$ to $x_i'$.

### 4.2 Computational Complexity

At first, projecting each point from $\mathbb{R}^d$ to a point in $\mathbb{D}_c^d$ has a complexity $\mathcal{O}(d)$ and projecting all points takes a complexity of order $\mathcal{O}(nd)$. As noted earlier, the estimation step of the root using a bootstrap sample of size $p'$ has a complexity of order $\mathcal{O}((p')^3 t)$. Translating each point again with respect to the position of the root takes another $\mathcal{O}(nd)$. The spectral decomposition has a complexity of order $\mathcal{O}(f^3)$ and as per our recommendation, by taking $f \approx \mathcal{O}(\log n)$, this step has a complexity of $\mathcal{O}((\log n)^3)$. Again, the normalization of rows and the following $k-$ means the clustering step takes an order of $\mathcal{O}(f^2 p)$. The final cluster assignment step has a complexity of order $\mathcal{O}(nfd) = \mathcal{O}(nd \log n)$. Combining all these, the final complexity of SHSC **Algorithm 2** is in the order of $\mathcal{O}((p')^3 t + nd \log n) + \mathcal{O}((\log n)^3) = \mathcal{O}((p')^3 t + nd \log n)$.

## 5 Theoretical Analyses

In this section, we will discuss the consistency of our SHSC **Algorithm 2**. Note that the analysis consists of two parts: First, we have to check the consistency associated with the estimation of the root and then the consistency of the subsequent scalable spectral clustering. We will start with the consistency of estimating the location of the root. We refer to Appendix A for all the proofs in this section.

---

**Algorithm 2** Scalable Hyperbolic Spectral Clustering Algorithm (SHSC)

---

**Input:** Dataset $\mathcal{X} := \{x_1, x_2, ..., x_n\} \in \mathbb{R}^d$, number of clusters=$p$, hyperparameter $\sigma$, cut-off length=$\epsilon$.

**Output:** Cluster labels $\mathcal{C} := \{C_1, C_2, ..., C_k\}$ where $C_i := \{j | x_j \in C_i\}$.

1: Obtain the transformed data points $\mathcal{X}' := \{x'_1, x'_2, ..., x'_n\} \in \mathbb{D}_c^d$ such that $x'_i := \exp_0^c(x_i)$.
2: Compute an estimate of the root node as per Algorithm 1, call it $r \in \mathbb{D}_c^d$.
3: Translate the set of points $x'_i$ with respect to $r$, i.e. obtain $y'_i := -r \oplus_c x'_i$. Let $\mathcal{Y}' := \{y'_1, y'_2, ..., y'_n\}$.
4: Perform a scalable spectral clustering as follows: Randomly sample $f(\approx \mathcal{O}(\log n))$ points from $\mathcal{Y}'$.
5: Construct the similarity matrix $W \in \mathbb{R}^{f \times f}$ with $W(i,j) := \begin{cases} exp\left(-\frac{d(y'_i, y'_j)^2}{\sigma^2}\right), & \text{if } d(y'_i, y'_j) \leq \epsilon \\ 0, & \text{otherwise.} \end{cases}$
6: Construct the diagonal degree matrix $D \in \mathbb{R}^{f \times f}$ with $D(i,j) := \begin{cases} \sum_{j=1}^N W(i,j), & \text{if } i = j \\ 0, & \text{otherwise.} \end{cases}$ .
7: Construct the Normalized Graph Laplacian, i.e. obtain $L := D - W \in \mathbb{R}^{f \times f}$. Then Construct $\tilde{L} := D^{-1/2} L D^{-1/2} \in \mathbb{R}^{f \times f}$.
8: Spectral Decomposition of the Normalized Graph Laplacian, i.e. obtain the first $p$ eigenvalues of $\tilde{L}$, $0 = \lambda_1 \leq \lambda_2 \leq ... \leq \lambda_p$ and the corresponding eigenvectors $u_i \in \mathbb{R}^f$, for $i \in \{1, 2, ..., p\}$. Let $U := [u_1, u_2, ...u_p] \in \mathbb{R}^{f \times p}$.
9: Normalize the rows of $U$, obtain $T \in \mathbb{R}^{f \times p}$ such that $T(i,j) := \frac{U(i,j)}{\sqrt{\sum_{l=1}^p U(i,l)^2}}$.
10: Representative Points on $\mathbb{R}^p$: Let $\mathcal{Z} := \{z_1, z_2, ..., z_f\} \in \mathbb{R}^p$, where $z_i$ represents $y'_i$ for $i \in \{1, 2, ..., p\}$ and $z_i^j = T(i,j)$.
11: Cluster Formation: Obtain the clusters $C_1, C_2, ..., C_k$ by performing $k-$ means clustering on $\mathcal{Z}$, where $C_i := \{j | z_j \in C_i\}$, i.e. $C_i := \{j | y_j \in C_i\}$.
12: Assign $x'_i$ to the cluster $C_j$ if the nearest neighbour of $x'_i$ in $\mathcal{Y}'$ is also in $C_j$, for $i \in \{1, 2, ..., n\}$ and $j \in \{1, 2, ..., p\}$.

---

### 5.1 Consistency of Estimating The Root

According to **Algorithm 1**, we have estimated the location of the root by considering the Fréchet Centroid or Fréchet Mean of the points corresponding to the lowest $K$ GHIs. For a complete metric space $(\mathcal{X}, d)$ and for points $\{x_1, x_2, ..., x_n\} \in \mathcal{X}$, one defines the Fréchet Variance of these points with respect to a point $p$ in $\mathcal{X}$ as

$$Var_F(p) := \sum_{i=1}^n d^2(p, x_i). \tag{9}$$

The Krachér Means are the minimizers of Expression in equation 9, and it is proved in Nielsen & Bhatia (2013) that if this minimizer is unique, then it is the Fréchet Mean of $\{x_1, x_2, ..., x_n\}$ with respect to the geometry of $\mathcal{X}$.

Note that if we draw a sequence of random variables $\{x_n\}_{n \geq 1}$ from $(\mathcal{X}, d)$ according to a probability distribution $P$ with the corresponding empirical distributions $P_n$, then the barycenter of $\mathcal{X}$ with respect to $P$, $\beta_P$ is defined as a minimizer of the function

$$x \mapsto \int_{\mathcal{X}} d^2(x, z) P(dz) \tag{10}$$

or in the empirical sense, the minimizers of the expression

$$x \mapsto \sum_{i=1}^n d^2(x_i, x) P_n(\{x_i\}), \tag{11}$$

where $P_n(\{x_i\}) = \frac{1}{n} \forall i \in \{1, 2, ..., n\}$. This notion of minimizers in 10 or 11 coincides with the notion of minimizers defined in 9.

Moreover, for a CAT(0) Space $(\mathcal{X}, d)$, any probability measure $P$ admits a unique barycenter with a concatenation property, the distance between two barycenters of two probability distributions $P$ and $Q$ is bounded by their $L^1-$ Wasserstein Distance, i.e. $d(\beta_P, \beta_Q) \leq W^1(P, Q)$ [see Theorem 4.7, Sturm, Sturm (2003)]. Although a more generalized version of Sturm's Theorem Sturm (2003) can be proved for a general $\delta-$ hyperbolic space [see Theorem 6.1, Ohta (2024)], for our purpose, Sturm's Theorem would be sufficient.

Having said all these, we are finally set to state the following theorem, which will provide the necessary stability (convergence of the empirical roots to the generalization root in probability) of the estimated root as mentioned in **Algorithm 1**, in a weak sense.

**Theorem 5.1.** *Let $\{X_i\}_{i \geq 1}$ be a sequence of independent, identically distributed random variables drawn from a CAT(0) Space $(\mathcal{X}, d)$ according to a probability distribution $P$ and let $\{P_i\}_{i \geq 1}$ be the corresponding empirical distributions. If $\{\beta_{P_i}\}_{i \geq 1}$ and $\beta_P$ are the corresponding empirical barycenters and the barycenter with respect to $P$ respectively, then*

$$d(\beta_{P_n}, \beta_P) \xrightarrow{P} 0 \ as \ n \to \infty. \tag{12}$$

**Remark 5.1.** Theorem 5.1 shows that the empirical barycenters converge to the generalized barycenter (true estimate) in probability. But according to our earlier comments, we are estimating the barycenters as Fréchet Centroids in **Algorithm 1**. Therefore, our estimate of the Fréchet Centroid will converge to the true estimate of the barycenter in probability as well.

### 5.2 Consistency of the Hyperbolic Spectral Decomposition

Now we talk about the consistency associated with the spectral decomposition of the hyperbolic similarity matrix. For this purpose, we will assume the participation of the entire dataset, i.e. we will omit the scalable part of our **Algorithm 2**, and will consider all points to construct the corresponding similarity metric. Following our previous notations, $x$ and $y$ are two points on the Poincaré ball of curvature $-c$, then their distance is given as

$$d(x, y) := 2 \sinh^{-1}\left(\sqrt{\frac{\delta_c(x, y)}{2}}\right).$$

where

$$\delta_c(x, y) = 2 \frac{\|x - y\|^2}{c(\frac{1}{c} - \|x\|^2)(\frac{1}{c} - \|y\|^2)}.$$

The Hyperbolic Gaussian Kernel $K_{H_G}$ is given as

$$K_{H_G}(x, y) = \exp(-a d(x, y)^2), a > 0.$$

Before going into the consistency analysis, we will look at a couple of results involved in the proof.

**Lemma 1.** *For the usual Euclidean Gaussian Kernel given by $K(x, y) = \exp(-a\|x - y\|^2)$ [treating $x, y$ as points in $\mathbb{R}^d$], we have $K_{H_G}(x, y) \leq K(x, y)$ whenever $x, y \in \mathbb{D}_c^d$.*

**Remark 5.2.** $K_{H_G}$ is radial: If we fix one variable, let's say $y$ at 0, then $\delta_c(x, 0) = 2\frac{\|x\|^2}{1/c - \|x\|^2}$, which is a radial function. Therefore, the metric 2 is also radial, and so is the Hyperbolic Gaussian Kernel.

**Lemma 2.** *The hyperbolic Gaussian Kernel $K_{H_G} \in L^1(H)$, i.e. this kernel is absolutely integrable.*

**Terminology.** For a compact subset $\Omega \in \mathbb{R}^d$ [with 0 in its interior], we call $\Omega$ to be symmetric if for every $x \in \Omega$ and for every $M \in SO_d(\mathbb{R}^d)$ [The group of all rotation matrices on $\mathbb{R}^d$], we have $Mx \in \Omega$.

**Lemma 3.** *Suppose $\Omega \in \mathbb{R}^d$ is symmetric, $f \in L^1(\Omega)$ and $f$ is radial. Then, its Fourier Transform is also radial.*

Next, we intend to use Theorem 3 Zhou (2002) and this necessitates computing the Fourier Transform $\widehat{K}(w)$ of $K_{H_G}(x)$ and will show that $\widehat{K}$ decays exponentially.

**Lemma 4.** *There exist $C, l > 0$ such that $\widehat{K}(w) \leq C \exp(-l\|w\|)$ for all $w \in \mathbb{R}^n$.*

**Terminology and Definitions.** Let $H$ be the compact subset of the Poincaré ball as defined above. $k : H \times H \to \mathbb{R}$ be the similarity function with the Gaussian Kernel equipped with the Poincaré Metric 2. Let $h : H \times H \to \mathbb{R}$ be the normalized similarity function. Then for any continuous function $g \in \mathcal{C}(H)$, we define the following [as in section 6 Von Luxburg et al. (2008):

$$
\begin{aligned}
\mathcal{K} &:= \{k(x, \cdot) : x \in H\}, \\
\mathcal{H} &:= \{h(x, \cdot) : x \in H\}, \\
g \cdot \mathcal{H} &:= \{g(x)h(x, \cdot) : x \in H\}, \\
\text{and} \, \mathcal{H} \cdot \mathcal{H} &:= \{h(x, \cdot)h(x, \cdot), x \in H\}.
\end{aligned}
$$

We also define $\mathcal{F} := \mathcal{K} \cup (g \cdot \mathcal{H}) \cup (\mathcal{H} \cdot \mathcal{H})$. We will re-write Theorem 19 Von Luxburg et al. (2008) with a slightly modified proof.

**Theorem 5.2.** *Let $(H, \mathcal{A}, P)$ be a probability space with $\mathcal{A}$ being any arbitrary sigma-algebra on $H$. Let $\mathcal{F}$ be defined as above with $\|f\|_\infty \leq 1$ for all $f \in \mathcal{F}$. Let $X_n$ be a sequence of i.i.d. random variables drawn according to the distribution $P$ and $P_n$ be the corresponding empirical distributions. Then there exists a constant $w > 0$ such that for all $n \in \mathbb{N}$ with probability at least $\delta$,*

$$
\sup_{f \in \mathcal{F}} |P_n f - P f| \leq \frac{w}{\sqrt{n}} \int_0^\infty \sqrt{\log(\mathcal{N}, \epsilon, L^2(P_n))} d\epsilon + \sqrt{\frac{1}{2n} \log\left(\frac{2}{\delta}\right)},
$$

*where $\mathcal{N}$ is the covering number of the space $H$ with ball of radius $\epsilon$ with respect to the metric $L^2(P_n)$. Hence the rate of convergence of the **Hyperbolic Spectral Clustering** is $\mathcal{O}\left(\frac{1}{\sqrt{n}}\right)$.*

**Remark 5.3.** Note that in deriving the convergence rate of the hyperbolic spectral clustering, we used results mostly to prove the consistency of spectral clustering in the Euclidean set-up. The hyperbolic metric is generally compelling compared to the squared Euclidean metric, which forces the hyperbolic Gaussian/Poisson Kernel to converge to 0 much faster than the Euclidean ones. Therefore, we believe the convergence rate of the hyperbolic spectral clustering can be improved, which requires estimating a careful bound on the logarithmic covering number with respect to the hyperbolic metric.

# 6 Experiments & Results

Previously we motivated the efficiency of our SHSC algorithm on the HTRU2 dataset in Section 4.1. Now, to evaluate the clustering performance of the proposed *Scalable Hyperbolic Spectral Clustering* (SHSC) algorithm on data with intrinsic hierarchical or categorical structure, we conduct experiments on a total of 5 large-scale real-world datasets. These datasets span multiple modalities, including lexical ontologies, text corpora, and image datasets, and are summarized in Table 1. Specifically, we consider the WordNet noun hierarchy Miller (1995), the DBpedia ontology hierarchy Lehmann et al. (2015), the Web of Science (WOS) hierarchical text classification dataset Kowsari et al. (2017), as well as the image datasets CIFAR-10 Krizhevsky (2009) and Fashion-MNIST Xiao et al. (2017), the latter two having flat categorical labels.

For text-based datasets, samples are represented using either TF–IDF features or pretrained language model embeddings, while for image datasets we use raw pixel values or deep convolutional features extracted from pretrained CNNs. Ground-truth hierarchical information, such as taxonomic depth or multi-level category labels, is used solely for evaluation and not during training or clustering.

We compare SHSC [with $p' = 100$, $K = 50$ and $c = 1.0$ in 1] against six representative hierarchical and deep clustering baselines:

1. *Hierarchical Deep Embedded Clustering* (HDEC) Shin et al. (2020), which extends deep embedded clustering to multi-level hierarchies in Euclidean latent space;

2. *SpectralNet* Shaham et al. (2018), a scalable deep spectral clustering method ;

| Dataset | Samples | Input Modality | Input Dimension | Clusters / Depth |
|---|---|---|---|---|
| WordNet (Noun Hierarchy) | $\sim 100k$ | Text (Glosses) | $\sim$2k (TF–IDF) | $5 \rightarrow 12$ |
| DBpedia Hierarchy | 342,782 | Text (Articles) | 300–1024 | $9 \rightarrow 70 \rightarrow 219$ |
| Web of Science (WOS) | $\sim 65k$ | Text (Abstracts) | 300–1024 | $21 \rightarrow 250 \rightarrow 4000$ |
| CIFAR-10 | 60,000 | Images (raw pixels) | $3 \times 32 \times 32$ | 10 classes |
| Fashion-MNIST | 70,000 | Images (raw pixels) | $28 \times 28$ | 10 classes |

Table 1: Large-scale hierarchical datasets used for evaluating Scalable Hyperbolic Spectral Clustering (SHSC). Input dimensionality depends on the chosen feature representation.

| Method | WordNet NMI(%) | DBpedia NMI (%) | WOS NMI (%) | CIFAR-10 NMI (%) | Fashion MNIST NMI (%) |
|---|---|---|---|---|---|
| HDEC | $61.47 \pm 2.45$ | $76.8 \pm 1.73$ | $79.85 \pm 2.07$ | $84.13 \pm 1.89$ | $68.57 \pm 1.72$ |
| SpectralNet | $84.38 \pm 2.76$ | $75.86 \pm 0.59$ | $82.35 \pm 1.20$ | $80.01 \pm 2.38$ | $54.26 \pm 2.93$ |
| DEC + Hierarchical | $71.29 \pm 1.74$ | $83.27 \pm 2.61$ | $78.51 \pm 1.42$ | $82.60 \pm 1.07$ | $63.65 \pm 0.43$ |
| HVAE (Hyperbolic VAE) | $84.21 \pm 2.89$ | $\mathbf{90.65 \pm 2.46}$ | $\mathbf{85.31 \pm 2.11}$ | $\mathbf{92.46 \pm 1.76}$ | $87.47 \pm 0.37$ |
| Poincaré + Clustering | $\mathbf{89.37 \pm 2.84}$ | $85.94 \pm 1.04$ | $81.59 \pm 1.37$ | $\underline{92.37 \pm 1.79}$ | $\underline{88.87 \pm 3.02}$ |
| Ward's | $77.45 \pm 1.27$ | $71.43 \pm 1.84$ | $74.57 \pm 2.39$ | $81.92 \pm 0.89$ | $73.82 \pm 2.43$ |
| **SHSC (Ours)** | $\underline{86.47 \pm 2.38}$ | $\underline{87.42 \pm 0.94}$ | $\underline{84.57 \pm 1.05}$ | $91.67 \pm 1.71$ | $\mathbf{92.35 \pm 1.82}$ |

Table 2: Normalized Mutual Information (NMI) comparison on hierarchical datasets. Means and standard deviations are computed over 25 runs.

3. *Deep Embedded Clustering* (DEC) Xie et al. (2016) combined with agglomerative hierarchical clustering;

4. *Hyperbolic Variational Autoencoders* (HVAE) Mathieu et al. (2019), which learn continuous hierarchical latent representations;

5. *Poincaré Embeddings* with post-hoc hierarchical clustering Nickel & Kiela (2017);

6. Classical Agglomerative hierarchical clustering with Ward linkage Ward (1963).

These baselines collectively cover Euclidean and hyperbolic geometries, spectral and embedding-based objectives, as well as shallow and deep hierarchical clustering paradigms.

All methods are evaluated using hierarchy-aware metrics, including normalized mutual information (NMI) computed at different hierarchy levels for the WordNet dataset, and standard NMI for flat-class datasets. Additional metrics include ancestor overlap consistency Bello et al. (2019); Ghosh et al. (2020), global hierarchy fidelity Bateni et al. (2024), and depth-sensitive hierarchical F1 scores Kosmopoulos et al. (2015); Kiritchenko & Matwin (2005) where applicable. The Python implementation of SHSC, along with data preprocessing and evaluation scripts, is publicly available at https://anonymous.4open.science/r/SHSC-CFFA/README.md.

## 6.1 Hierarchy-Aware Evaluation: Ancestor Overlap

To quantify the effectiveness of clustering methods in capturing hierarchical relationships, we employ the *Ancestor Overlap* (AO) metric Bello et al. (2019); Ghosh et al. (2020). Ancestor Overlap measures the fraction of correctly preserved parent-child relationships between predicted clusters and the ground-truth hierarchy. Higher values indicate a stronger alignment of the clustering output with the underlying taxonomic structure. This metric is particularly informative for datasets such as WordNet, DBpedia, and the Web of Science (WOS), which possess multi-level hierarchical organization Mikolov et al. (2013); Nickel & Kiela (2017); Murtagh & Contreras (2017). Unlike standard NMI, which evaluates clustering quality at a single level Xie et al. (2016), AO explicitly penalizes violations of the hierarchy, ensuring that both coarse and fine-grained structures are respected.

As shown in Table 3, Euclidean-based methods such as HDEC and classical agglomerative clustering with Ward linkage achieve moderate AO scores, reflecting their limited capacity to model complex hierarchical

dependencies. SpectralNet improves performance on WordNet and WOS by leveraging a spectral objective to capture global structure but underperforms on the deeper DBpedia hierarchy Shaham et al. (2018). DEC combined with agglomerative refinement offers noticeable gains, indicating that hierarchical post-processing benefits deep embeddings Xie et al. (2016). Hyperbolic methods, including Poincaré embeddings and Hyperbolic VAEs, naturally encode hierarchical relationships in their latent space, leading to higher ancestor overlap Nickel & Kiela (2017); Mathieu et al. (2019). Importantly, our proposed SHSC algorithm attains top performance on DBpedia and WOS (88.43% and 83.85%, respectively) compared to nearest competitive counterparts like HVAE and Poincaré+Clustering, while maintaining competitive results on WordNet (86.92%). These results demonstrate that SHSC effectively preserves hierarchical parent-child relationships across diverse datasets, corroborating the trends observed in the NMI evaluation and confirming the advantage of combining hyperbolic representation with scalable spectral clustering for hierarchical data Ghosh et al. (2020).

| Method | WordNet
Ancestor Overlap (%) | DBpedia
Ancestor Overlap (%) | WOS
Ancestor Overlap (%) |
|---|---|---|---|
| HDEC | $68.28 \pm 2.17$ | $75.60 \pm 1.82$ | $71.37 \pm 2.05$ |
| SpectralNet | $81.55 \pm 2.52$ | $72.48 \pm 1.10$ | $78.27 \pm 1.39$ |
| DEC + Hierarchical | $74.38 \pm 1.92$ | $79.81 \pm 2.29$ | $73.62 \pm 1.47$ |
| HVAE | $83.28 \pm 2.67$ | $\underline{87.17 \pm 2.95}$ | $\underline{82.34 \pm 1.72}$ |
| Poincaré + Clustering | $\mathbf{88.71 \pm 2.80}$ | $84.53 \pm 1.37$ | $80.94 \pm 1.57$ |
| Ward's | $76.19 \pm 1.35$ | $70.50 \pm 1.63$ | $74.28 \pm 2.19$ |
| **SHSC (Ours)** | $\underline{86.92 \pm 2.27}$ | $\mathbf{88.43 \pm 1.04}$ | $\mathbf{83.85 \pm 2.46}$ |

Table 3: Hierarchy-aware Ancestor Overlap comparison. Higher values indicate better preservation of hierarchical structure.

### 6.1.1 Statistical Significance of Ancestor Overlap Improvements

To determine whether the gains of our SHSC algorithm are statistically significant and not due to random variability, we conducted pairwise *two-sample t-tests* comparing SHSC against each competing method. For each method, we used the AO scores obtained across multiple independent runs ($n = 25$) and performed a one-sided t-test with the null hypothesis that SHSC does not outperform the baseline:

$$H_0 : \mu_{\text{SHSC}} \leq \mu_{\text{baseline}}, \quad H_1 : \mu_{\text{SHSC}} > \mu_{\text{baseline}}.$$

The t-statistic was computed as

$$t = \frac{\bar{x}_{\text{SHSC}} - \bar{x}_{\text{baseline}}}{\sqrt{\frac{s_{\text{SHSC}}^2}{n} + \frac{s_{\text{baseline}}^2}{n}}},$$

where $\bar{x}$ and $s$ denote the mean and standard deviation of AO scores for the respective methods. Degrees of freedom were approximated using Welch's formula.

Table 4 reports the one-sided t-test p-values comparing SHSC against each baseline across the three hierarchical datasets. Except for a few cases, such as Poincaré + Clustering, most improvements achieved by SHSC are statistically significant.

These results confirm that the observed improvements of SHSC over most baselines, particularly Euclidean and spectral methods, are highly unlikely to be due to noise. In particular, the gains on WordNet and DBpedia are consistently significant, while hyperbolic baselines such as Poincaré embeddings show comparable performance on some datasets, reflecting the strong hierarchical modeling capability shared by hyperbolic methods under structured geometry.

| Baseline Method | WordNet | DBpedia | WOS |
|---|---|---|---|
| HDEC | $< 0.001$ | $< 0.001$ | $< 0.001$ |
| SpectralNet | $< 0.001$ | $< 0.001$ | $< 0.001$ |
| DEC + Hierarchical | $< 0.001$ | $< 0.001$ | $< 0.001$ |
| HVAE | $< 0.001$ | $0.009$ | $0.008$ |
| Poincaré + Clustering | $1.000$ | $< 0.001$ | $< 0.001$ |
| Ward's | $< 0.001$ | $< 0.001$ | $< 0.001$ |

Table 4: One-sided t-test p-values comparing SHSC to each baseline across three hierarchical datasets using Ancestor Overlap Metric. Values below 0.05 indicate statistically significant improvement of SHSC.

## 6.2 Global Hierarchy Fidelity: Dendrogram Purity

We further evaluate *Dendrogram Purity* in order to examine how well different methods preserve hierarchical class coherence across all levels of the learned tree, as reported in Table 5. Dendrogram Purity measures the extent to which samples sharing the same ground-truth label are merged early in the clustering hierarchy Bateni et al. (2024). Specifically, for each pair of points belonging to the same class, the purity of the smallest subtree containing both points is computed, and the final score is obtained by averaging over all such pairs. Unlike flat metrics such as NMI, this measure explicitly evaluates the global structure of the dendrogram and penalizes premature or incorrect merges Marco & Marín (2007), making it particularly suitable for hierarchical datasets such as WordNet, DBpedia, and WOS, while remaining informative for induced hierarchies on image datasets.

| Method | WordNet Purity (%) | DBpedia Purity (%) | WOS Purity (%) | CIFAR-10 Purity (%) | Fashion-MNIST Purity (%) |
|---|---|---|---|---|---|
| HDEC | $72.15 \pm 1.84$ | $78.42 \pm 1.67$ | $75.31 \pm 1.92$ | $85.46 \pm 1.28$ | $70.62 \pm 1.74$ |
| SpectralNet | $83.76 \pm 2.31$ | $77.58 \pm 1.02$ | $80.47 \pm 1.41$ | $82.39 \pm 2.10$ | $58.74 \pm 2.66$ |
| DEC + Hierarchical | $76.94 \pm 1.59$ | $82.11 \pm 2.05$ | $77.63 \pm 1.26$ | $84.28 \pm 1.01$ | $66.41 \pm 0.88$ |
| HVAE | $85.62 \pm 2.43$ | $\underline{88.94 \pm 2.18}$ | $84.72 \pm 1.95$ | $\mathbf{93.41 \pm 1.52}$ | $89.11 \pm 0.64$ |
| Poincaré + Clustering | $\mathbf{89.85 \pm 2.54}$ | $86.27 \pm 1.19$ | $82.36 \pm 1.38$ | $\underline{93.18 \pm 1.65}$ | $\underline{90.24 \pm 2.71}$ |
| Ward's | $79.21 \pm 1.18$ | $73.64 \pm 1.52$ | $76.83 \pm 2.07$ | $83.91 \pm 0.83$ | $75.29 \pm 2.31$ |
| **SHSC (Ours)** | $\underline{88.94 \pm 2.11}$ | $\mathbf{89.76 \pm 0.91}$ | $\mathbf{86.05 \pm 1.08}$ | $92.84 \pm 1.60$ | $\mathbf{93.18 \pm 1.47}$ |

Table 5: Dendrogram Purity comparison. Higher values indicate better preservation of hierarchical structure.

| Baseline Method | WordNet | DBpedia | WOS | CIFAR-10 | Fashion-MNIST |
|---|---|---|---|---|---|
| HDEC | $< 0.001$ | $< 0.001$ | $< 0.001$ | $< 0.001$ | $< 0.001$ |
| SpectralNet | $< 0.001$ | $< 0.001$ | $< 0.001$ | $< 0.001$ | $< 0.001$ |
| DEC + Hierarchical | $< 0.001$ | $< 0.001$ | $< 0.001$ | $< 0.001$ | $< 0.001$ |
| HVAE | $< 0.001$ | $0.142$ | $< 0.001$ | $1.000$ | $< 0.001$ |
| Poincaré + Clustering | $1.000$ | $< 0.001$ | $< 0.001$ | $1.000$ | $< 0.001$ |
| Ward's | $< 0.001$ | $< 0.001$ | $< 0.001$ | $< 0.001$ | $< 0.001$ |

Table 6: One-sided t-test p-values comparing SHSC to each baseline across five datasets for the dendrogram Purity metric. Values below 0.05 indicate statistically significant improvement of SHSC.

As shown in Table 5, Euclidean-based methods such as HDEC and Ward's linkage achieve moderate purity scores Tichỳ et al. (2010), indicating limited ability to preserve hierarchical coherence at deeper levels. SpectralNet improves upon these baselines by capturing global similarity structure but remains constrained by its Euclidean embedding. Hyperbolic approaches, including Poincaré embeddings and Hyperbolic VAEs, consistently yield higher dendrogram purity, confirming the effectiveness of negatively curved spaces for modeling hierarchical data. Notably, the proposed *Scalable Hyperbolic Spectral Clustering* (SHSC) algorithm achieves the highest performance on DBpedia, WOS, and Fashion-MNIST, while staying competitive on

WordNet and CIFAR-10. These results demonstrate that SHSC effectively preserves global hierarchical structure, complementing the gains observed in NMI and ancestor-based evaluations and validating its design for scalable hierarchical clustering.

Similar to the evaluation on Ancestor Overlap, we performed one-sided t-tests for the dendrogram Purity metric to assess whether the observed gains are due to random variability or are statistically significant. As shown in Table 6, except for a few cases such as Poincaré + Clustering, most of the improvements achieved by SHSC are statistically significant, with corresponding p-values below 0.05.

### 6.3 Hierarchy-aware Evaluation using Hierarchical F1 Score

We report the *Hierarchical F1 (hF1)* scores in Table 7 to assess the hierarchy-preserving quality of different clustering methods. Hierarchical F1 accounts for the closeness of predicted clusters to the true hierarchical structure by giving partial credit when predictions lie along the correct path in the hierarchy Kosmopoulos et al. (2015); Kiritchenko & Matwin (2005).

Across the DBpedia and WOS datasets, our proposed SHSC method consistently achieves the higher hF1 scores, while maintaining comparable performance on WordNet. This highlights its effectiveness in preserving hierarchical relationships. Methods based on hyperbolic embeddings, such as Poincaré + Clustering, also perform well, reflecting the suitability of hyperbolic geometry for modeling tree-like structures Nickel & Kiela (2017). In contrast, traditional Euclidean clustering approaches, including HDEC and Ward's, exhibit lower hF1 scores, indicating their limitations in capturing multi-level hierarchical information.

Overall, the hierarchical F1 metric complements Ancestor Overlap by quantifying not only whether ancestor-descendant relations are preserved, but also how closely predicted clusters align with the true hierarchical paths, providing a more accurate evaluation of hierarchy-aware clustering.

Following the evaluation on Ancestor Overlap and Dendrogram Purity, we conducted one-sided t-tests on the hF1 metric to determine whether the observed improvements are statistically significant. As reported in Table 8, with the exception of a few cases such as Poincaré + Clustering, the gains achieved by SHSC are significant, with p-values below 0.05.

| Method | WordNet Hierarchical F1 (%) | DBpedia Hierarchical F1 (%) | WOS Hierarchical F1 (%) |
|---|---|---|---|
| HDEC | $66.15 \pm 2.05$ | $73.42 \pm 1.76$ | $69.18 \pm 1.98$ |
| SpectralNet | $79.03 \pm 2.44$ | $70.95 \pm 1.05$ | $76.08 \pm 1.31$ |
| DEC + Hierarchical | $72.10 \pm 1.85$ | $77.62 \pm 2.20$ | $71.05 \pm 1.40$ |
| HVAE | $81.12 \pm 2.55$ | $\underline{85.71 \pm 2.85}$ | $\underline{79.80 \pm 1.65}$ |
| Poincaré + Clustering | $\mathbf{86.45 \pm 2.70}$ | $82.97 \pm 1.30$ | $78.35 \pm 1.50$ |
| Ward's | $74.05 \pm 1.28$ | $68.93 \pm 1.58$ | $72.45 \pm 2.10$ |
| **SHSC (Ours)** | $\underline{84.21 \pm 2.15}$ | $\mathbf{86.77 \pm 1.00}$ | $\mathbf{81.93 \pm 2.35}$ |

Table 7: Hierarchy-aware evaluation using Hierarchical F1. Higher values indicate better alignment of predicted clusters with the true hierarchy.

| Baseline Method | WordNet | DBpedia | WOS |
|---|---|---|---|
| HDEC | $< 0.001$ | $< 0.001$ | $< 0.001$ |
| SpectralNet | $< 0.001$ | $< 0.001$ | $< 0.001$ |
| DEC + Hierarchical | $< 0.001$ | $< 0.001$ | $< 0.001$ |
| HVAE | $< 0.001$ | $0.019$ | $< 0.001$ |
| Poincaré + Clustering | $1.000$ | $< 0.001$ | $< 0.001$ |
| Ward's | $< 0.001$ | $< 0.001$ | $< 0.001$ |

Table 8: One-sided t-test p-values comparing SHSC to each baseline method across the three hierarchical datasets using Hierarchical F1. Values below 0.05 indicate statistically significant improvement of SHSC.

### 6.4 Depth-Sensitive Evaluation on WordNet: Level-wise NMI

To further analyze how well different methods capture hierarchical structure at varying levels of granularity, we report *level-wise Normalized Mutual Information (NMI)* on the WordNet noun hierarchy in Table 9Mikolov et al. (2013); Nickel & Kiela (2017). In this evaluation, clustering performance is assessed independently at multiple depths of the hierarchy, corresponding to increasingly fine-grained semantic distinctions. Level 1 represents coarse semantic groupings near the root of the taxonomy, while Levels 2 and 3 correspond to progressively deeper and more specialized categories. This depth-sensitive evaluation provides a more nuanced understanding of hierarchical clustering performance than a single flat NMI score.

As shown in Table 9, all methods achieve relatively high NMI at Level 1, indicating that coarse semantic distinctions are easier to recover across competing methods. However, performance degrades at deeper hierarchies, particularly for Euclidean-based methods such as HDEC and Ward's linkage, reflecting their limited ability to represent fine-grained hierarchical structure Murtagh & Contreras (2017). SpectralNet and DEC with hierarchical post-processing offer moderate improvements, benefiting from global structure modeling and agglomerative refinement. Hyperbolic approaches Shaham et al. (2018); Xie et al. (2016), including Poincaré embeddings and Hyperbolic VAEs, consistently outperform Euclidean baselines, confirming the suitability of hyperbolic geometry for modeling hierarchical data. Notably, the proposed SHSC algorithm achieves higher NMI at Levels 2 and 3 (83.77% and 75.40%, respectively), while remaining competitive at Level 1. These results demonstrate that SHSC not only preserves coarse hierarchical organization but also excels at capturing deeper semantic distinctions, highlighting its effectiveness in modeling complex, multi-level hierarchies Nickel & Kiela (2017); Mathieu et al. (2019).

| Method | WordNet Level 1 | WordNet Level 2 | WordNet Level 3 |
|---|---|---|---|
| HDEC | $85.21 \pm 2.02$ | $72.56 \pm 1.889$ | $61.73 \pm 1.36$ |
| SpectralNet | $88.74 \pm 2.40$ | $77.36 \pm 1.57$ | $66.91 \pm 1.90$ |
| DEC + Hierarchical | $87.10 \pm 1.05$ | $75.87 \pm 0.98$ | $64.25 \pm 1.62$ |
| HVAE | $90.17 \pm 2.53$ | $80.79 \pm 1.85$ | $70.94 \pm 1.78$ |
| Poincaré + Clustering | $\mathbf{93.22 \pm 2.86}$ | $\underline{82.93 \pm 1.37}$ | $\underline{71.56 \pm 1.03}$ |
| Ward's | $74.64 \pm 1.51$ | $71.20 \pm 1.86$ | $60.37 \pm 2.59$ |
| **SHSC (Ours)** | $\underline{91.52 \pm 2.27}$ | $\mathbf{83.77 \pm 1.24}$ | $\mathbf{75.40 \pm 1.54}$ |

Table 9: Level-wise NMI on the WordNet noun hierarchy. SHSC consistently achieves higher NMI at deeper levels, demonstrating superior capacity to capture hierarchical semantics.

| Baseline Method | WordNet Level 1 | WordNet Level 2 | WordNet Level 3 |
|---|---|---|---|
| HDEC | $< 0.001$ | $< 0.001$ | $< 0.001$ |
| SpectralNet | $< 0.001$ | $< 0.001$ | $< 0.001$ |
| DEC + Hierarchical | $< 0.001$ | $< 0.001$ | $< 0.001$ |
| HVAE | $0.011$ | $< 0.001$ | $< 0.001$ |
| Poincaré + Clustering | $1.000$ | $0.006$ | $< 0.001$ |
| Ward's | $< 0.001$ | $< 0.001$ | $< 0.001$ |

Table 10: One-sided t-test p-values comparing SHSC to each baseline at different levels of the WordNet hierarchy using NMI. Values below 0.05 indicate statistically significant improvement of SHSC.

Similar to the previous evaluations, we conducted one-sided t-tests on the level-wise NMI scores to assess whether the observed improvements of SHSC are statistically significant. As shown in Table 10, with the exception of a few cases such as Poincaré+Clustering or HVAE, most improvements are significant, with corresponding p-values below 0.05.

### 6.5 Computational Efficiency and Scalability

In addition to clustering quality, practical deployment of hierarchical clustering algorithms crucially depends on empirical computational efficiency. To this end, we report the average wall-clock runtime Yan et al.

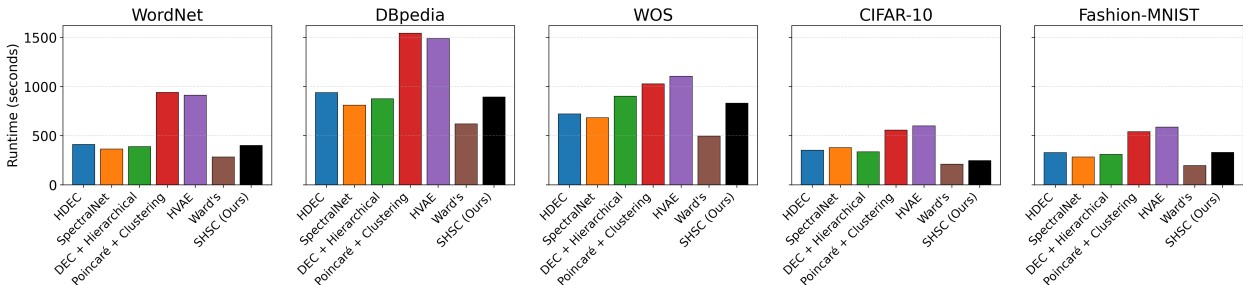

Figure 4: Empirical runtime comparison (in seconds) across five benchmarks datasets. Each subplot corresponds to one dataset, with bars denoting different clustering methods using a consistent color scheme across datasets. SHSC demonstrates favorable computational efficiency relative to competing hierarchical and non-Euclidean baselines.

(2009) per clustering run (in seconds) for all competing methods across five datasets of varying scale and modality, summarized in Figure 4. All methods are evaluated under identical experimental conditions, using the same hardware configuration [an HP Envy $x360$ laptop with an NVIDIA RTX 3050 GPU, Intel $i7$ CPU, and 16 GB RAM] and stopping criteria. This metric captures the end-to-end cost of representation learning and hierarchical clustering, thereby reflecting real-world usability rather than theoretical computational complexity 4.2.

As expected, classical agglomerative approaches such as Ward's linkage exhibit the lowest runtime due to their simplicity, but this comes at the expense of reduced hierarchical fidelity, as we observed in previous sections. Deep Euclidean methods (HDEC, DEC + Hierarchical, and SpectralNet) incur additional computational overhead from representation learning, while hyperbolic methods based on Poincaré embeddings and Hyperbolic Variational Autoencoders show substantially higher runtimes, particularly on large-scale text datasets such as DBpedia and WOS, specifically due to expensive Riemannian optimization Bonnabel (2013). Notably, **SHSC** achieves a favorable balance between efficiency and performance, due to it's inherent non-deep nature: it consistently remains superior compared to deep hyperbolic baselines in runtime while remaining competitive with Euclidean methods. These results demonstrate that SHSC scales effectively across both vision and text datasets, making it a practical choice for large-scale hierarchical clustering on hyperbolic space without compromising structural accuracy.

### 6.6 Ablation Studies

We conduct an ablation study to examine the sensitivity of the proposed method with respect to three hyperparameters: (i) the number of bootstrap samples $p'$ used to robustly estimate the latent root node, (ii) the number of selected nodes $K$ with the lowest Gromov hyperbolicity indices, and (iii) the curvature parameter $c$ of the hyperbolic embedding space. Clustering performance is evaluated using Normalized Mutual Information (NMI) on the Fashion-MNIST dataset. The results are shown in Figure 5.

**Effect of the number of bootstrap samples $p'$.** The parameter $p'$ controls the number of bootstrap samples used to locate the root node based on the $K$ lowest Gromov hyperbolicity indices. As $p'$ increases from 50 to 175, the NMI curves become smoother and exhibit reduced variance across different values of $K$, which strongly aligns with our consistency analyses in 5.1, showing that the estimated root node probabilistically converges to the true root node as the number of bootstrap samples becomes very large, which in turn stabilizes downstream clustering performance.

**Effect of the number of selected lowest $K$ GHIs.** Across all configurations, the NMI remains relatively stable as $K$ varies between 40 and 70. This robustness suggests that the method does not require precise tuning of $K$, provided that the selected nodes correspond to low Gromov hyperbolicity values.

**Effect of the curvature parameter $c$.** We evaluate four curvature values $c \in \{0.001, 1, 10, 100\}$, corresponding to increasingly negative curvature in the hyperbolic space. Moderate curvature values ($c = 1$ and $c = 10$) consistently yield strong and stable NMI scores. In contrast, very small curvature ($c = 0.001$), which approaches a near-Euclidean regime, exhibits higher variability, while excessively large curvature ($c = 100$)

can lead to mild performance degradation due to geometric distortion. These observations highlight the importance of selecting an appropriate and moderate curvature selection for hierarchical representation.

This ablation study demonstrates that reliable root estimation benefits from a sufficient number of bootstrap samples, that the method is robust to the choice of low-hyperbolicity nodes, and that moderate hyperbolic curvature provides the best trade-off between optimal performance and stability.

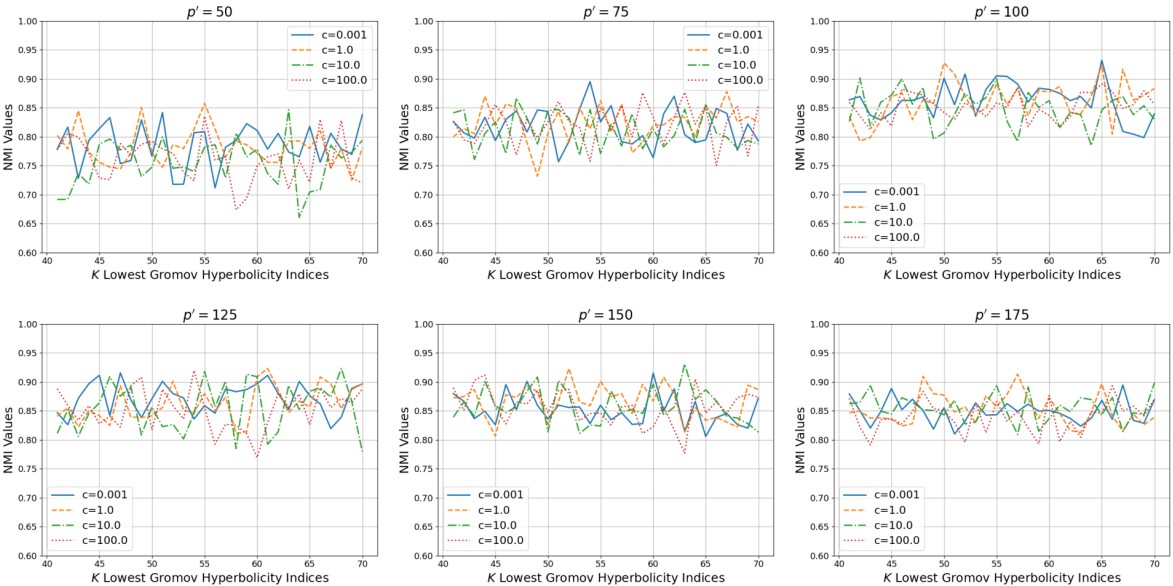

Figure 5: Ablation study analyzing the effect of the number of bootstrap samples $p'$ used for root estimation, the number $K$ with the lowest Gromov hyperbolicity indices, and the curvature parameter $c$ of the Poincaré ball on clustering performance on the Fashion-MNIST dataset measured by NMI. Each subplot corresponds to a different value of $p'$, while curves within each subplot show results for varying curvature values. The results demonstrate robustness to the choice of $K$, improved stability with increasing $p'$, and optimal performance under moderate curvature.

## 6.7   A Robust Estimation of the curvature parameter

To select the curvature parameter in a data-driven manner, we estimate it from the empirical Gromov hyperbolicity of the dataset. Specifically, we repeatedly draw bootstrap samples of size $p'$ and compute the Gromov Hyperbolicity Indices (GHIs) for sampled quadruples using the four-point condition. From the pooled set of GHIs across all bootstrap iterations, we retain the lowest $K$ values, which correspond to the most tree like local configurations in the data. The median of these values, denoted by $\hat{\delta}$, provides a robust estimate of the intrinsic hyperbolicity of the dataset, the algorithm has been described in 3. The curvature parameter is then determined as $c = 1/\hat{\delta}^2$, which increases the magnitude of negative curvature for datasets exhibiting stronger tree like structures.

A limitation of this procedure is that the estimating the curvature depends on bootstrap sampling and the choice of $K$, which may introduce additional variability in the curvature estimate. Moreover, since the GHIs are computed from local triplets [Equation 8], the resulting curvature primarily reflects local hyperbolic structure and may not fully capture global geometric heterogeneity present in complex datasets. Nevertheless, this approach provides a simple and computationally feasible mechanism for adapting the curvature of the Poincaré ball to the intrinsic geometry of the data.

---

**Algorithm 3** Curvature Estimation using Lowest $K$ Triplet Hyperbolicity Scores

---

**Input:** Dataset $\mathcal{X} := \{x_1, x_2, ..., x_n\} \in \mathbb{R}^d$

**Hyperparameters:** Bootstrap Sample Size $= p'$, Bootstrap Iterations $= t$, Number of lowest scores $= K$

**Output:** Curvature parameter $c$

1: Initialize an empty list $\mathcal{D}$ to store triplet hyperbolicity scores
2: **for** $i = 1$ to $t$ **do**
3:     Draw a bootstrap sample $\mathcal{X}^{(i)} \subset \mathcal{X}$ of size $p'$
4:     Compute pairwise distances between points in $\mathcal{X}^{(i)}$
5:     **for** each triplet $(a, b, c)$ in $\mathcal{X}^{(i)}$ **do**
6:         Compute

$$\delta(a, b, c) = \frac{1}{2}\Big(d(a, b) + d(a, c) - d(b, c)\Big)$$

7:         Append $\delta(a, b, c)$ to $\mathcal{D}$
8:     **end for**
9: **end for**
10: Sort $\mathcal{D}$ in ascending order
11: Extract the lowest $K$ values: $\mathcal{D}_K$
12: Compute the median score $\hat{\delta} = \text{median}(\mathcal{D}_K)$
13: Estimate curvature

$$c = \frac{1}{\hat{\delta}^2}$$

14: **return** $c$

---

### 6.8 Stability of Nearest neighbor Assignment in Algorithm 2 and in Algorithm 5

An additional practical consideration revolves around the stability of nearest-neighbor assignments on the Poincaré Ball, for points located far from the root, which is present both in Algorithm 2 [Step 12], and Algorithm 5 [Step 1]. In the Poincaré ball model, hyperbolic distances grow exponentially as points approach the boundary of the ball. As a consequence, points that lie farther from the root tend to exhibit increasingly larger separations in their pairwise hyperbolic distances. This exponential divergence makes nearest-neighbor relationships more stable under small perturbations of the embedding, since relative distance differences between candidate neighbors become more pronounced. Therefore, the nearest-neighbor assignment procedure used in constructing the similarity matrix is generally robust for peripheral nodes in the hierarchy. However, when points lie extremely close to the boundary, small numerical perturbations may still be amplified, which can introduce minor variability in the resulting neighborhood structure.

## 7 Limitations of the Single-Root Assumption and an Extension to Multi-Root SHSC

Our proposed SHSC algorithm relies on estimating a single latent root 1 on the Poincaré ball, which serves as a reference point for translating the embedded data via Möbius addition. This formulation is motivated in particular by the observation that many hierarchical datasets, such as taxonomies or tree or graph like datasets, admit a dominant root hierarchy. By relocating the estimated root to the origin, SHSC normalizes the hyperbolic representation and enables stable spectral decomposition on the resulting similarity graph 2.

However, this single root assumption may be restrictive for datasets whose underlying structure is not governed by a single global hierarchy. In practice, many real-world datasets exhibit multiple loosely connected hierarchies or several branching regions that cannot be well represented by a single latent node. In such situations, forcing all points to be expressed relative to one root may introduce distortions in the radial translation of the hyperbolic embedding, potentially affecting the quality of the affinity matrix used in the spectral clustering stage.

A natural extension of the SHSC framework is, therefore, to consider a *multi-root* formulation. Instead of estimating a single Fréchet mean centroid from the lowest Gromov hyperbolicity triplets, one could identify multiple candidate roots corresponding to different low-hyperbolicity regions of the dataset. Each point can then be assigned to its nearest latent roots and can be translated accordingly through Möbius addition with respect to that nearest root in the metric of the Poincaré Ball, instead of translating all the points relative to a single latent node. This effectively produces a collection of locally centered hyperbolic coordinate systems, allowing the algorithm to capture multiple hierarchical substructures within the same dataset.

Such a multi-root strategy may better accommodate datasets with heterogeneous or multi-scale hierarchical organization, while preserving the core formulation of SHSC. We now discuss a strategy to simultaneously estimate multiple nodes (*latent roots*) using the Gromov Hyperbolicity Indices which further extends the algorithm proposed in 1.

## 7.1 m Roots Estimaton using Top K Gromov Hyperbolicity Indices.

---

**Algorithm 4** $m$ Roots Estimation using lowest $K$ Gromov Hyperbolicity Indices

---

**Input:** Dataset $\mathcal{X} := \{x_1, x_2, ..., x_n\} \in \mathbb{R}^d$

**Hyperparameters:** Bootstrap Sample Size $= p'$, Bootstrap Iterations $= t$, Number of lowest GHIs $= K$, Number of latent roots $= m$ $[mK << \binom{p'}{3}]$, Curvature $= -c\,(c > 0)$ of the Poincaré ball.

**Output:** Estimates of $m$ root nodes

1: Obtain the transformed set of points $\mathcal{X}' := \{x_1', x_2', ..., x_n'\} \in \mathbb{D}_c^d$ such that $x_i' := \exp_0^c(x_i)$.
2: **for** iterations$= 1, 2, ..., t$ **do**
3:     pick a bootstarp sample of size $p'$, i.e. a random subset $\mathcal{Y} \subseteq \mathcal{X}'$ of size $p'$.
4:     for each 3 combinations of points of $\mathcal{Y}$, calculate the GHIs.
5:     select the lowest $mK$ GHIs and their corresponding 3 points for each of those GHIs. Let's call these points $\{pt_1, pt_2, \ldots, pt_{3mK}\}$ stored in a way such that for all $j \in \{1, 2, \ldots, mK\}$, the triplet $\{pt_{3(j-1)+1}, pt_{3(j-1)+2}, pt_{3(j-1)+3}\}$ correspond to the $j$-th lowest GHI.
6: **end for**
7: **for** $i = 1, 2, \ldots, m$ **do**
8:     estimate the $i$th root $r_i$ as the Fréchet mean of $\{pt_{(i-1)K+1}, \ldots, pt_{(iK)}\}$
9: **end for**

---

Once we estimate those $m$ roots, the next stage is to translate all the datapoints relative to their nearest root. This can be easily done through the Algorithm 5

---

**Algorithm 5** Nearest Neighbor Assignment and Local Translation

---

**Input:** Dataset $\mathcal{X} := \{x_1, x_2, ..., x_n\} \in \mathbb{R}^d$, $m$ roots $\mathcal{M} = \{r_1, \ldots, r_m\}$

**Hyperparameters:** Curvature $= -c\,(c > 0)$ of the Poincaré ball.

**Output:** Translated points $x_i' = x_i \oplus_c (-r_{j(i)})$

1: For each $i \in \{1, 2, \ldots, n\}$, let $r_{j(i)}$ be the nearest latent root of $x_i$ with respect to the Poincaré metric equation 2, for $j \in \{1, 2, \ldots, m\}$
2: **for** iterations$i = 1, 2, \ldots, n$ **do**
3:     Compute the translated points $x_i' = x_i \oplus_c (-r_{j(i)})$
4: **end for**

---

The translated points can then be used as input to Algorithm 2 for hyperbolic spectral clustering. Since the local translations have already been performed, steps 2 and 3 of Algorithm 2 are no longer required.

## 7.2 Experiments with Multi-Root SHSC

We now evaluate the clustering performance of the proposed multi-root SHSC algorithm on the WordNet and DBpedia datasets. In particular, we implement the multi-root SHSC procedure described in Algorithm 4, using the same hyperparameters $p' = 100$, $K = 50$, and curvature $c = 1.0$. The experiments are conducted for five different values of the number of latent roots, $m \in \{1, 2, 3, 4, 5\}$.

For each setting of $m$, we assess clustering performance using three hierarchical evaluation metrics: Ancestor Overlap (AO), Dendrogram Purity (DP), and Hierarchical F1-score (HF1). To ensure comparability, we follow exactly the same experimental setup as in the single-root experiments described previously, with the only difference being the number of estimated latent roots $m$.

| Evaluation Metric | $m = 1$ | $m = 2$ | $m = 3$ | $m = 4$ | $m = 5$ |
|---|---|---|---|---|---|
| Ancestor Overlap (%) | $86.92 \pm 2.27$ | $87.88 \pm 3.43$ | $87.24 \pm 3.62$ | $86.25 \pm 2.86$ | $87.12 \pm 4.01$ |
| Dendogram Purity (%) | $88.94 \pm 2.11$ | $88.35 \pm 2.79$ | $87.36 \pm 4.54$ | $88.78 \pm 3.82$ | $86.02 \pm 4.13$ |
| Hierarchical F1 (%) | $84.21 \pm 2.15$ | $85.71 \pm 2.89$ | $86.13 \pm 3.74$ | $85.04 \pm 3.92$ | $85.93 \pm 4.35$ |

Table 11: Performance of Multi-Root SHSC on **WordNet** dataset across different number of estimated roots

| Evaluation Metric | $m = 1$ | $m = 2$ | $m = 3$ | $m = 4$ | $m = 5$ |
|---|---|---|---|---|---|
| Ancestor Overlap (%) | $88.43 \pm 1.04$ | $87.92 \pm 2.93$ | $88.01 \pm 2.78$ | $87.92 \pm 3.42$ | $87.46 \pm 3.25$ |
| Dendogram Purity (%) | $89.76 \pm 0.91$ | $90.04 \pm 1.67$ | $90.34 \pm 2.39$ | $89.76 \pm 3.07$ | $88.94 \pm 2.97$ |
| Hierarchical F1 (%) | $86.77 \pm 1.00$ | $85.34 \pm 2.09$ | $87.49 \pm 1.84$ | $87.39 \pm 2.19$ | $86.28 \pm 2.95$ |

Table 12: Performance of Multi-Root SHSC on **DBpedia** dataset across different number of estimated roots

## 7.3 Discussions.

Tables 11 and 12 summarize the clustering performance of the multi-root SHSC algorithm on the WordNet and DBpedia datasets for varying numbers of estimated roots $m \in \{1, 2, 3, 4, 5\}$. While the single-root case ($m = 1$) provides more stable and competitive performance across all metrics, introducing multiple roots leads to increased variability in the results. For both datasets, we observe that the metrics fluctuate as $m$ increases, without any clear trend of improvement.

These observations suggest that implementing the multi-root SHSC algorithm introduces inherent geometric distortions in the embedding space. When bootstrapping samples to identify the lowest $mK$ Gromov Hyperbolicity Indices (GHIs) for determining the locations of $m$ roots, the resulting GHIs across the bands [Step 8, Algorithm 4] can vary across iterations. This leads to inconsistent root estimations. By applying local translations around multiple roots, we are effectively imposing different local geometric structures. Since the local GHI of a root reflects the hyperbolicity of its neighborhood, these translations distort the uniform negative curvature of the underlying hyperbolic space.

Although all points remain on the same Poincaré ball, the local distortions create regions with effectively varying curvature across neighborhoods. This phenomenon explains the inconsistencies observed in the multiple-root estimation procedure described in Algorithm 4. Under this setup, the consistency guarantees provided in Theorem 5.1 no longer hold, since there is no global uniform geometric structure. Consequently, the increased variability observed in Tables 11 and 12 can be attributed to these geometric inconsistencies.

While there are occasional marginal improvements when using multiple roots, the heightened variability prevents us from confidently attributing these gains to true structural capture, rather than noise introduced by local distortions in the Poincaré ball.

These observations suggest that naively increasing the number of roots does not automatically yield better hierarchical clustering. Proper selection of multiple roots and their local influence is likely crucial to maintain stability and improve performance. This highlights an important direction for future work: developing principled strategies for multi-root selection and translation that balance local hierarchical representation with global consistency.

# 8 Conclusion & Future Work

In this paper, we addressed the challenges of representing hierarchical and tree-like data in low-dimensional Euclidean spaces, highlighting their limitations in preserving hierarchical distances and relationships. We demonstrated that hyperbolic spaces provide a natural and efficient alternative, enabling accurate embeddings even in low dimensions, and leading to competitive hierarchy-preserving clustering performance, as confirmed by experiments on WordNet, DBpedia, WOS, CIFAR-10 and Fashion-MNIST datasets.

Our primary contribution is a scalable spectral clustering algorithm designed specifically for the Poincaré ball. The method replaces the conventional Euclidean similarity matrix with a hyperbolic similarity matrix, constructed after estimating the hierarchy's latent root and translating the dataset relative to it via Möbius addition. Theoretical analysis establishes the weak consistency of the proposed method, with convergence rates comparable to standard spectral clustering in Euclidean spaces. Overall, these results confirm that leveraging hyperbolic geometry can significantly enhance hierarchical clustering while maintaining computational scalability.

While the current SHSC algorithm assumes a single latent root, our discussion in Section 7 highlights that extending the method to multiple roots introduces geometric distortions and increased variability in evaluation metrics. These observations suggest that naive multi-root implementations can compromise the uniform hyperbolic structure of the embedding space. Developing principled strategies for selecting multiple roots and appropriately translating local neighborhoods presents an important direction for future work. We anticipate that such extensions could further improve the algorithm's ability to capture complex hierarchical structures, particularly in datasets containing multiple loosely connected hierarchies, while preserving stability and consistency.

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

## A    Appendix

**Theorem 5.1**  Let $\{X_i\}_{i \geq 1}$ be a sequence of independent, identically distributed random variables drawn from a CAT(0) Space $(\mathcal{X}, d)$ according to a probability distribution $P$ and let $\{P_i\}_{i \geq 1}$ be the corresponding empirical distributions. If $\{\beta_{P_i}\}_{i \geq 1}$ and $\beta_P$ are the corresponding empirical barycenters and the barycenter with respect to $P$ respectively, then

$$d(\beta_{P_n}, \beta_P) \xrightarrow{P} 0 \text{ as } n \to \infty. \tag{13}$$

*Proof.* Shorack and Wellner [Shorack & Wellner (2009)] showed that the Kantorovich or $L^1-$ Wasserstein distance between two probability measures is exactly equal to the $L^1$ distance between their cumulative distributions, i.e. if $\{F_n\}_{n \geq 1}$ are the empirical cumulative distribution functions and $F$ is the cumulative distribution corresponding to $P$, then

$$W^1(P_n, P) = \|F_n - F\|_{L^1(\mathcal{X}, d)}. \tag{14}$$

Moreover, Sturm [Sturm (2003)] showed that $d(\beta_{P_n}, \beta_P) \leq W^1(P_n, P)$. Combining this with Equation 14, we get

$$d(\beta_{P_n}, \beta_P) \leq W^1(P_n, P) = \|F_n - F\|_{L^1(\mathcal{X}, d)}. \tag{15}$$

But $\|F_n - F\|_{L^1(\mathcal{X}, d)} \to 0$ as $n \to \infty$ by dominated convergence theorem. Hence, $d(\beta_{P_n}, \beta_P) \xrightarrow{P} 0$ as $n \to \infty$ by 15 as well, completing the proof of Theorem 5.1.

$\square$

**Lemma 1.**    For the usual Euclidean Gaussian Kernel given by $K(x, y) = \exp(-a\|x - y\|^2)$, we have $K_{H_G}(x, y) \leq K(x, y)$ whenever $x, y \in \mathbb{D}_c^d$.

*Proof.* **Step 1:** We have $\|x\|, \|y\| < 1/\sqrt{c}$, hence $\|x\|^2, \|y\|^2 < 1/c \implies (1/c - \|x\|^2)(1/c - \|y\|^2) < 1/c^2$. Hence we can write

$$\frac{\|x - y\|^2}{c(1/c - \|x\|^2)(1/c - \|y\|^2)} > c\|x - y\|^2.$$

But we also have

$$\delta_c(x, y) = 2 \frac{\|x - y\|^2}{c(\frac{1}{c} - \|x\|^2)(\frac{1}{c} - \|y\|^2)}.$$

Combining this with the last inequality, we get

$$\delta_c(x, y) > 2c\|x - y\|^2.$$

**Step 2:** For $x \in \mathbb{R}$, $\frac{d}{dx}(\sinh^{-1}(x)) = \frac{1}{\sqrt{1+x^2}} > 0$. Therefore the inverse sine hyperbolic function is a strictly increasing function of $x$. By Step 1, we have $\delta_c(x, y) > 2c\|x - y\|^2$. Therefore, we have $\frac{\delta_c(x,y)}{2} \geq c\|x - y\|^2$. This also implies $\sqrt{\frac{\delta_c(x,y)}{2}} \geq \sqrt{c}\|x - y\|$.

Since $d(x, y) = 2\sinh^{-1}\left(\sqrt{\frac{\delta(x,y)}{2}}\right)$ and the inverse sine hyperbolic function is increasing, we can write

$$d(x, y) \geq 2\sinh^{-1}(\sqrt{c}\|x - y\|)$$

We know that for $0 < s < t$, $\exp(-s) > \exp(-t)$. This enables us to write

$$K_{H_G}(x,y) = \exp(-ad(x,y)^2)$$
$$\leq \exp(-4a[\sinh^{-1}(c\|x-y\|)]^2).$$

**Step 3:** Note that for $0 \leq x \leq 1$, $\frac{1}{\sqrt{1+x^2}} \geq \frac{1}{\sqrt{2}}$. Let $f(x) := \sinh^{-1}(x) - \frac{x}{2}$. Then $f$ is differentiable and we get $f'(x) = \frac{1}{\sqrt{1+x^2}} - \frac{1}{2} \geq \frac{2-\sqrt{2}}{2\sqrt{2}}$. Therefore $f$ is increasing on $[0,1]$ and for $0 \leq x \leq 1$, $\sinh^{-1}(x) \geq \frac{x}{2}$. Hence $\exp(-\sinh^{-1}(\|x-y\|^2)) \leq \exp\left(-\frac{\|x-y\|^2}{2}\right)$. Therefore following step 2, we get

$$K_{H_G}(x,y) \leq \exp(-4a[\sinh^{-1}(c\|x-y\|)]^2) \leq \exp\left(-4ac\frac{\|x-y\|^2}{4}\right) = \exp(-ac\|x-y\|^2) = K(x,y),$$

$\square$

**Lemma 2.** The hyperbolic Gaussian Kernel $K_{H_G} \in L^1(H)$, i.e. this kernel is absolutely integrable.

*Proof.* $K_{H_G}(x) = K_H(x,0) \leq K(x,0) = \exp(-ac\|x\|^2)$ [by Lemma 1 and Remark 5.2]. Therefore following step 3 of Lemma 1 we write,

$$\int_H |K_{H_G}(x)|dx \leq \int_H |\exp(-ac\|x\|^2)|dx = \int_H \exp(-ac\|x\|^2)dx \leq \int_{\mathbb{R}^n} \exp(-ac\|x\|^2)dx < \infty.$$

as $H$ is any compact subset of $\mathbb{D}_c^d$, we can also think of $H$ as an embedded as a subset of the ball of radius $1/\sqrt{c}$ embedded in $\mathbb{R}^d$, with the Euclidean metric replaced by 2. The last integral is finite since the integrand is the usual Gaussian distribution. $\square$

**Lemma 3.** Suppose $\Omega \in \mathbb{R}^d$ is symmetric, $f \in L^1(\Omega)$ and $f$ is radial. Then, its Fourier Transform is also radial (Grafakos & Teschl, 2013).

*Proof.* For a function $f : \Omega \subset \mathbb{R}^d \to \mathbb{R}$, the Fourier Transform is defined as $f \mapsto \int_\Omega f(x)e^{-i\langle w,x\rangle}dx$. $f$ is radial if and only if for every $M \in SO_d(\mathbb{R}^d)$ [where $SO_d(\mathbb{R}^d)$ is the special unitary group on $\mathbb{R}^d$, i.e. consisting of all $d \times d$ matrices over $\mathbb{R}$ with determinant 1], $f(Mx) = f(x)$ [as the operation $x \to Mx$ only rotates $x$, does not change its magnitude, i.e. $\|Mx\| = \|x\|$]. Then for any arbitrary $M \in SO_d(\mathbb{R}^d)$,

$$\widehat{f}(Mt) = \int_\Omega f(x)e^{-i<Mt,x>}dx$$
$$= \int_{M(\Omega)} f(Ms)e^{-i\langle Mt,Ms\rangle}ds \quad \text{[change of variable } x \to Ms\text{]}$$
$$= \int_\Omega f(s)e^{-i\langle t,s\rangle}ds \quad \text{[since } \Omega \text{ is symmetric]}$$
$$= \widehat{f}(t),$$

where the second equality follows from the conjugate linearity of the inner product: $\langle Mt, Ms\rangle = \langle M^*Mt, s\rangle = \langle t,s\rangle$ since $M^*M = I_d$ [$M \in SO_d(\mathbb{R}^d)$]. $\square$

**Lemma 4.** There exist $C, l > 0$ such that $\widehat{K}(w) \leq C\exp(-l\|w\|)$ for all $w \in \mathbb{R}^n$.

*Proof.* Let $f(x) = K_{H_G}(x) = exp(-acd(x,0)^2)$. Then by Lemma 1, we have $f(x) \leq exp(-ac\|x\|^2)$ for all $x \in H$. Exploiting the fact that $\widehat{K}$ is radial (and hence real-valued), we get

$$
\begin{aligned}
|\widehat{K}(w)| = \left| \int_H f(x) e^{-iw^t x} dx \right| &= \left| \int_H f(x) e^{ac\|x\|^2} e^{-ac\|x\|^2} e^{-iw^t x} dx \right| \\
&\leq \int_H |f(x) e^{ac\|x\|^2} e^{-ac\|x\|^2} e^{-iw^t x}| dx \\
&\leq \int_H |e^{-ac\|x\|^2} e^{-iw^t x}| dx \\
&\leq \int_{\mathbb{R}^n} |e^{-a\|x\|^2} e^{-iw^t x}| dx \\
&\leq C' e^{-p\|w\|^2} \\
&\leq C exp(-l\|w\|),
\end{aligned}
$$

where the second inequality is followd by noting that $\int_H |e^{-ac\|x\|^2} e^{-iw^t x}| dx$ is the Fourier Transform of the Euclidean gaussian kernel over $H$. where $C'$ and $C$ are some appropriately chosen constants. □

**Theorem 5.2** Let $(H, \mathcal{A}, P)$ be a probability space with $\mathcal{A}$ being any arbitrary sigma-algebra on $H$. Let $\mathcal{F}$ be defined as above with $\|f\|_\infty \leq 1$ for all $f \in \mathcal{F}$. Let $X_n$ be a sequence of i.i.d. random variables drawn according to the distribution $P$ and $P_n$ be the corresponding empirical distributions. Then there exists a constant $w > 0$ such that for all $n \in \mathbb{N}$ with probability at least $\delta$,

$$
\sup_{f \in \mathcal{F}} |P_n f - P f| \leq \frac{w}{\sqrt{n}} \int_0^\infty \sqrt{\log(\mathcal{N}, \epsilon, L^2(P_n))} d\epsilon + \sqrt{\frac{1}{2n} \log\left(\frac{2}{\delta}\right)},
$$

where $\mathcal{N}$ is the covering number of the space $H$ with ball of radius $\epsilon$ with respect to the metric $L^2(P_n)$. Hence **the rate of convergence of the Hyperbolic Spectral Clustering is $\mathcal{O}\left(\frac{1}{\sqrt{n}}\right)$.**

*Proof.* Combining Lemma 4 and Theorem 3 Zhou (2002) we get

$$
\log(\mathcal{N}(\mathcal{F}, \epsilon, \| \cdot \|_\infty)) \leq C_0 \log\left(\frac{1}{\epsilon}\right)^{d+1},
$$

for some constant $C_0$ chosen appropriately and $d$ is the dimension of $H$. Since $d$ is a constant for $H$, we can write the above inequality as

$$
\log(\mathcal{N}(\mathcal{F}, \epsilon, \| \cdot \|_\infty)) \leq C_1 \left(\log\frac{1}{\epsilon}\right)^2.
$$

Following the same sequence of computation as in Theorem 19 Von Luxburg et al. (2008), we get

$$
\int_0^\infty \sqrt{\log(\mathcal{N}(\mathcal{F}, \epsilon, L^2(P_n)))} d\epsilon < \infty
$$

Hence following Theorem 19 Von Luxburg et al. (2008) we write

$$
\sup_{f \in \mathcal{F}} |P_n f - P f| \leq \frac{w}{\sqrt{n}} \int_0^\infty \sqrt{\log(\mathcal{N}(\mathcal{F}, \epsilon, L^2(P_n)))} d\epsilon + \sqrt{\frac{1}{2n} \log\left(\frac{2}{\delta}\right)} < \frac{C_1}{\sqrt{n}} + \sqrt{\frac{1}{2n} \log\left(\frac{2}{\delta}\right)},
$$

for some appropriately chosen constant $C_1$. Since $\delta > 0$ we get,

$$
\sup_{f \in \mathcal{F}} |P_n f - P f| \leq C\left(\frac{1}{\sqrt{n}}\right).
$$

Finally, combining theorem 16 of Von Luxburg et al. (2008) with the last inequality, we get

$$\sup_{f \in \mathcal{F}} |P_n f - P f| = \mathcal{O}\left(\frac{1}{\sqrt{n}}\right).$$

$\square$

