# OpenReview forum: "Consistent Spectral Clustering under Hyperbolic Geometry"
_TMLR — Rejected by TMLR_

### Review · Reviewer_hrqE · 2026-03-11

**Summary Of Contributions:**

This paper proposes Scalable Hyperbolic Spectral Clustering (SHSC), which is designed for data with hierarchical or tree-like structures. The authors argue that Euclidean geometry distorts these relationships. The method uses the Poincaré ball model and involves estimating a latent hierarchical root using Gromov Hyperbolicity Indices, followed by a spectral decomposition on a subset of points. Theoretical analysis is provided to show that the method is weakly consistent with convergence rates similar to classical spectral clustering. The authors evaluate the model on several large-scale datasets, including WordNet and DBpedia, using hierarchy-aware metrics like Ancestor Overlap and Dendrogram Purity.

**Audience:**

Yes

**Audience Explanation:**

This work will be of interest to some researchers in non-Euclidean representation learning and spectral clustering. Even if the empirical results are currently marginal, the demonstration that a scalable, non-deep method can stay competitive with deep learning models in structured settings is a valuable finding for some segment of the TMLR audience.

**Broader Impact Concerns:**

This is a methodological paper on clustering with no obvious societal risks. Standard comments on compute cost apply but a dedicated statement is not necessary.

**Claims And Evidence:**

No

**Claims Explanation:**

I do not believe the evidence fully supports the authors' claims of superiority over existing methods. The standard deviations across 25 runs are quite high, often in the 2-3% range, which makes it difficult to tell if the reported gains are actually meaningful. On datasets like DBpedia and WOS, the performance gap between SHSC and baselines like HVAE is often smaller than these margins of error. Furthermore, the results are inconsistent across benchmarks. In Table 2, SHSC actually underperforms relative to HVAE on DBpedia and relative to Poincare + Clustering on WordNet.

While the theoretical consistency proofs seem solid, the empirical narrative needs to be significantly adjusted. The evidence supports the method as a potentially competitive alternative, but it does not support the claim that it is a superior performer in terms of accuracy.

**Requested Changes:**

A list of my requested changes, as well as whether they are critical for securing my recommendation for acceptance, is given below:

1. Perform a rigorous statistical significance analysis, using t-tests to report p-values for the results in Tables 2, 3, 4, and 5. Given the high variance, the authors need to prove that these gains are not just noise. (critical for acceptance)

2. Tone down the empirical claims throughout the paper. The method should be presented as a competitive, theoretically-grounded alternative rather than a superior one where the error bars clearly overlap. (critical for acceptance)

3. Address the limitation of the single-root assumption in Algorithm 1. Many real hierarchies are multi-rooted and I would like to see a discussion or experiment on how the model handles these cases. (critical for acceptance)

4. Provide a more principled heuristic or method for selecting the curvature parameter c, as the ablation studies show the performance is quite sensitive to this choice. (recommended)

5. Include a brief discussion on the stability of the nearest-neighbor assignment for points far from the root where hyperbolic distances diverge exponentially. (recommended)

Corrections:

- In Table 2, HVAE should be bolded for the datasets DBPedia and WOS
- The text in Algorithms 1 and 2 is currently very small and hard to read. Please make the text larger and/or rewrite the algorithms to be more terse and clear
- Correct the capitalization in itemized item 6 on page 12 (e.g. "Classical agglomerative") and "Lemma 1" in the appendix

---

> ### Author Response · Authors · 2026-03-17
> **Response to Reviewer hrqE**
>
> We thank the reviewer for the valuable feedback and pertinent remarks. We have addressed the raised concerns in the following way. We also like to emphasize that all the relevant revisions are in blue in our revised manuscript.
>
> **Requested Changes**
>
> **R1 Statistical Test of Significance to justify gains in evaluation tables:**
> We thank the reviewer for this feedback. Throughout Sections 6.1, 6.2, 6.3 and 6.4, after each of the mentioned Tables we have performed a one-sided t-test and inserted the p-values to justify that in most of the cases, the performance gains achieved by our SHSC algorithm is not purely due to noise (except for a few cases like Poincar\’{e} Clustering). We expect this insertion to show that our SHSC algorithm remains competitive with modern deep and non-deep hierarchical clustering algorithms.
>
> **R2 Rephrasing the empirical claims made throughout the paper:**
> We sincerely apologize for the confusion regarding our statements of the empirical claims made throughout our paper, and we thank the reviewer for pointing out this discrepancy. Our main objective was to introduce a scalable non-deep clustering algorithm in a structured geometric setting that can provide competitive performance compared to modern deep and non-deep hierarchical clustering algorithms. We have now rephrased similar claims wherever it was mentioned earlier, and now state that our SHSC algorithm can provide a competitive performance, and at the same time, it can remain scalable and computationally efficient.
>
> **R3 Limitations of Single Root Assumption in Algorithm 1 and Introducing Multi-Root SHSC:**
> We are really grateful to the reviewer for this constructive feedback. We have now introduced a completely new Section, namely Section 7: "Limitations of the Single-Root Assumption and an Extension to Multi-Root SHSC", to discuss this topic. Additionally, we have introduced Algorithms 5 and 6 to estimate multiple roots and performed experiments over the Wordnet and DBpedia datasets when no of latent roots can vary from 1 to 5. We further refer to Sections 7.2 and 7.3 for the experimental results and discussions.
>
> **R4 A Principled heuristic to determine the curvature parameter:**
> We have introduced this heuristic in Section 6.7 as a principled approach to determine the curvature based on Gromov Hyperbolicity Indices.
>
> **R5 Stability of the Nearest Neighbour Assignment:**
>  We have included a brief discussion on this in Section 6.8
>
> **Corrections:**
>
> **In Table 2, HVAE should be bolded for the datasets DBPedia and WOS:**
> We have subsequently modified this in our revision.
>
> **The text in Algorithms 1 and 2 is currently very small and hard to read:**
> We have enlarged the text format in all the Algorithms in our revision and made it comparable to the original text of the manuscript.
>
> **The capitalization errors in itemized item 6 on page 12 (e.g. "Classical agglomerative") and "Lemma 1" in the appendix:**
> We have also rectified these capitalization errors in our revision.

---

### Review · Reviewer_JvRt · 2026-03-31

**Summary Of Contributions:**

This paper introduces an approach for discovering hierarchical structures using spectral clustering with a hyperbolic geometry. The paper presents an algorithm, SHSC, accompanied with theoretical and empirical results. The algorithm, as described by the authors, consists of three parts: 1. Root Estimation using lowest K Gromov Hyperbolicity Indices, 2. Performing a Scalable Spectral Clustering Algorithm and 3. Final Cluster label Assignment on a Nearest Neighbor Basis. Empirical results demonstrate that on several datasets, in terms of several metrics, the authors' method outperforms baseline approaches.

Strengths:
1. Empirical evidence - the authors show they can outperform baseline methods on most datasets
2. Section 7's observations and contributions are quite nice

Weaknesses:
Despite the clear revisions to make this more clear, I am still struggling with an understanding of how to think about these relatively high dimensional datasets whose labels are hierarchical in terms of the hyperbolic space. Section 7 makes some of the clearest arguments.
It feels more like the authors are finding nails for their given hammer with these datasets, rather than studying a scientific property. Figure 4 is perhaps most important here, but I feel it is presented in a way that leaves me feeling the paper is under developed.

In short, I think the weakness is that the claim of _why_ these are good evaluations of hyperbolic clustering could be better developed. I think a stronger paper storyline is one that says. Empirically, look at these datasets, they have labels that are clearly hyperbolic, and they have GHI that is indicative and now look at these other datasets where that is not true. Then show that normal spectral clustering works on the latter datasets but not the former. Then show how your method fixes this. Then show how important Section 7's observation is.

I think that a storyline that is more like this would better highlight the contributions of the paper.

Minor note: many references are malformed.

**Audience:**

Yes

**Audience Explanation:**

Yes, I think so. The work has some connections to folks interested in non-euclidean representations and clustering. I think the paper is more geared toward practitioners than theorists, in which case it would have been nice to see a more unique / specific application.

**Claims And Evidence:**

No

**Claims Explanation:**

My main concern is the storyline of the empirical results. I think this is the crux of the paper? I think the theoretical results are really just further tools for the technical understanding of the method (the theory itself is not the core contribution). I think the empirical storyline shows the success of the method, but doesn't tell the scientific story very clearly.

**Requested Changes:**

Please fix the malformed references. Please at least consider re-framing some of the (empirical) story as I mention above.

I mention this because I truly think this is a stronger story for the paper. I think it would make the paper more appreciated by the community. I also think that that it would guide the narrative even from the introduction much better. Then you can replace Figure 1 / Figure 2 with observations from real data, not these toy examples, which make the reader be concerned of the practical use of the method. I think it would also make the empirical results seem all the more impressive.

---

> ### Author Response · Authors · 2026-04-14
> **Response to Reviewer JvRt**
>
> We thank the reviewer for the valuable feedback and pertinent remarks. We have addressed the raised concerns in the following way. We also like to emphasize that all the relevant revisions are in dark green colour in our revised manuscript.
>
> **Requested Changes:**
>
> **R1 Please fix the malformed references.**
>
> Response:
> We apologise for these malformed references. This occurred due to the unintentional use of \citet{} instead of \citep{}. We have now fixed all of them.
>
>
>
> **R2 Please at least consider re-framing some of the (empirical) story as I mention above.**
>
> Response:
> We thank the reviewer for this feedback. We have reframed our empirical storyline perspective. We have now motivated the formulation of our SHSC algorithm through two well-known real-life datasets, one of which is the IRIS dataset, which has a high Gromov hyperbolicity index, where the Euclidean Spectral Clustering algorithm works well compared to our SHSC. In contrast, we have also experimented with the HTRU2 dataset, which has a relatively lower Gromov hyperbolicity index, where our SHSC works better compared to the conventional Euclidean Spectral Clustering. Although we motivated our proposed algorithm through two relatively smaller real datasets, while running the empirical evaluations, we benchmarked our method against the modern non-deep and deep hierarchical clustering algorithms on large, complex real-life datasets.

---

### Review · Reviewer_2dR8 · 2026-04-03

**Summary Of Contributions:**

The authors provide new version of spectral clustering for data lying along a hyperbolic manifold. The key to the method is embedding the data into a hyperbolic space (Poincare ball model), forming an affininty graph (constructed using hyperbolic distances), and then applying (scalable) spectral clustering. They show that their method works well on data sets with known hierarchical (and thus hyperbolic-ish) structure.

They also state two theorems which are intended to establish statistical consistency of their barycenter estimation (used to define the root of their trees, i.e., hierarchies) and of the hyperbolic spectral decomposition.

Overall, this paper introduces a reasonable, intuitive algorithm and shows it works well in the releveant context. However, there are major issues with the theory.

Addition there is a severe number of typos (to the point where it makes it hard to focus on the math) and this reads more like a rough draft than a submission.

**Additional Comments:**

Question:

Alg 4, why do you only consider {pt_{i-1}K+1,....pt_{ik}}? Wouldn't it make sense to consider earlier points that haven't been "picked" yet?

**Audience:**

Yes

**Audience Explanation:**

Hierarchical data appears in many real-world networks necessitating the need for data science methods to handle such data.

**Claims And Evidence:**

No

**Claims Explanation:**

There appears to an issue with the proof of the first theorem and the statement of the second

**Requested Changes:**

Major:

1. The proof of Theorem 5.1 is based on a result relating $W^1$ distances to CDF distances. However, in order for CDFs to make sense, the random variables must by $\mathbb{R}$-valued (or at least in some poset). However, $P$ and $P_n$ appear to be probability distributions for RVs taking values in $\mathbb{H}$ (or more generally in $\mathcal{X}$. In this case, this the CDFs don't make sense.

2. The statement of Theorem 5.2 does not make sense. L^2(P_n) is a metric on functions defined on $H$ but the result claims that N is the covering number of $H$. Also, Pf and P_nf are not defined, nor is their definition obvious. What does it mean for a probability distribution to act on a function?

3. Page 4: The authors claim that tangent spaces are one-dimensional manifolds. This is blatantly wrong and undermines my trust in the rest of the paper, particularly considering that I am not an expert on differential geometry and there are numerous other issues.

4. Lemma 1, step 3. It is unclear why $\|x-y\|^2\leq 1$. We only have that $\|x\|,\|y\|\leq 1$.

5. More detail should be given in the proof of Theorem 5.2 (assuming the statement can be made to make sense). How does the $\|\cdot\|_\infty$ change to an $L^2(P_n)$.

Minor:

1. Description of 2-means clustering in the intro is misleading. Typically in the 2-class case one simply uses the second eigenvector. The clusters are the points where it is positive/negative.

2. Throughout, very many places, the there are times parentheses are in the wrong spot in citations. Usually this can be fixed via careful use of \citet and \citep.

3. Page 2, first paragraph, curly braces missing in X := x_1,...

4. "to bipartite" should be "to partition" (same paragraph)

5. "l being the number of labels"... why are there labels if this is clustering. Unsupervised. Also, $\ell$ not "$l$". It helps it look less like a capital i.

6. Page 3, incorrect capitalization of "Graph" repeated.

7. Use \langle and \rangle in LaTeX for inner products

8. Page 6, missing citation for Rips

9. More background should be given on the adjusted rand index

10. Page 7, "discussed in 3" should be "discussed in Section 3." This mistake, and similar ones are repeated throughout the paper.

11. Page 8. Capitalize "therefore"

12. Page 8, X: {x_1,...,x_n} with the equals sign missing

13. Use \ll rather than << for much less than

14. Semi-major. Unclear if the Frechet mean will be one of the data points. Does this matter?

15. Alg 1. "bootstarp"

16. Page 9: Capitalize "translating"

17. Should refer to algorithms 1 and 2 more prominently in the main text.

18. Page 10. Incorrect capitalization of Random

19. Page 10. "minimizers of 9" should be "minimizers of (9)" (use \eqref not \ref).

20. Page 10. P_n(x_i) should be P_n(\{x_i\})

21. Page 11. Incorrect capitalization of Theorem.

22. Theorem 5.1. The definitions of $P_i$ should be more clearly stated to make it clear that it is the empirical distribution of data points up to i. Also, inconsistency with $P_n$ vs $P_i$.

23. The authors should give the definition of the Fourier transform since there are multiple "standard" definitions.

24. Page 12: Missing space after "and" in the Terminology and Definitions.

25. Page 13: "Aagglomerative"

26. In the tables, you should probably also underline the second best (and bold the best). This helps you look good.

27. Reporting runtimes is good, but ideally they should be in the same table as the performance results for ease of comparison by the reader.

28. Using $\|\cdot\|$ by itself to mean the hyperbolic metric is problematic and bad notation. This is usually reserved for the Euclidean metric.

29. Lemma 3 is correct, but "known" should probably cite an analysis textbook or something.

30. Weird \times symbol in the first line of the proof of lemma 4

31. Second line of proof of lemma four, lowercase k vs capital K.

32. Page 29, "Finally Finally"

---

> ### Author Response · Authors · 2026-04-17
> **Response to Reviewer 2dR8**
>
> We thank the reviewer for the valuable feedback and pertinent remarks. We have addressed the raised concerns in the following way. We also like to emphasize that all the relevant revisions are in red colour in our revised manuscript.
>
> **Requested Changes (major)**
>
> **R1** In high-dimensional probability theory, it is well known that the CDF of a multivariable function is well defined; we just need one additional condition to define the distribution function [for concrete citation, we refer to this work: "https://www.sciencedirect.com/science/article/pii/S0167947321001018". ].
>
> **R2** We do not claim that $ \mathcal{N} $ represents the covering number of $\mathbb{H} $ in Theorem 5.2. The proof of this theorem closely follows the methodology used in Theorem 19 of the well-known work by von Luxburg, "Consistency of Spectral Clustering, which we have explicitly cited multiple times throughout the manuscript. Moreover, the distinction between the actions of empirical and true probability measures on functions is standard and should be clear within the framework established in that reference.
>
> **R3** We sincerely apologize for this unintentional wording here; we have rectified this in our revision.
>
> **R4** In Lemma 1, Step 3, we mentioned that $0 < x < 1$; therefore, for two points $x,y$ both in $[0,1]$, it is obvious that $|x-y| \leq 1$.
>
> **R5** We have already indicated that the central argument of this proof follows the same technique as that used in Theorem 19 of von Luxburg’s work. However, to rigorously implement this approach, the preceding steps, namely, Lemmas 1 through 4, are essential.
>
>
>
> **Requested Changes (minor)**
>
> **R 2,3,4,5,6,7,8,10,11,12,13,15,16,17,18,19,20,21,24,25,29,30,31,32**
> We have corrected all the typos, misplaced citations, capitalization issues, and math symbols.
>
> **R1** We have modified this 2-class clustering in the introduction.
>
> **R9** We have now modified our motivating examples using real-life datasets based on other reviewers' feedback, where we have given a brief description of the Normalized Mutual Information (NMI).
>
> **R14** The Frechet mean could be one of the data points, in which case, one of the data points will act as the latent node in the hierarchy; it does not obstruct the main flow of the algorithm design.
>
> **R22** We have defined $\{P_n\}$ in Page 11.
>
> **R23** We defined the Fourier Transform on Page 29.
>
> **R26** We have underlined the second-best results in the Table as well.
>
> **R27** We noted the runtime for each algorithm to compute the class labels. Since each table represents a different metric, we prefer to keep the runtime separated from the tables.
>
> **R28** We have used the notation $|x-y|$ to indicate the distance between $x,y$ when treating $x,y$ as points in the Euclidean Space and used $d(x,y)$ to indicate the distance between $x,y$ on the Poincare Ball $\mathbb{D}_c^d$.

---

### Decision · Action_Editor_PaqR · 2026-05-13

**Recommendation:** Reject

**Additional Comments:**

While the application of hyperbolic geometry to spectral clustering addresses a highly relevant and interesting problem for the TMLR community, I must reject this submission due to critical, unresolved mathematical flaws in its theoretical analysis. As rigorously detailed by the reviewers, particularly Reviewer 2dR8, the proof establishing weak consistency (Theorem 5.1) invalidly relies on a 1-dimensional Euclidean identity—relating the Wasserstein distance to the $L_1$ distance between Cumulative Distribution Functions (CDFs)—that is not canonically defined or justified for random variables on a hyperbolic manifold. Furthermore, the statement of Theorem 5.2 retains an ill-defined mathematical formulation regarding the covering number of the manifold with respect to an empirical function metric ($L^2(P_n)$). Unfortunately, the authors dismissed these valid mathematical critiques in their rebuttal and failed to correct the underlying errors in the revised manuscript. Because the core theoretical claims are built on flawed mathematical foundations that remain unaddressed, the submission fails to meet TMLR's baseline criteria for claims being supported by accurate, convincing, and clear evidence.

**Audience:**

Yes

**Audience Explanation:**

The intersection of hyperbolic geometry and unsupervised learning is a growing and compelling area of machine learning research. Adapting foundational techniques like spectral clustering to better handle hierarchical or tree-like data by moving beyond Euclidean assumptions is a highly relevant problem for the TMLR community. While this specific manuscript suffers from fundamental theoretical flaws that preclude its acceptance, researchers working in manifold learning, graph representations, and clustering would certainly be interested in the underlying conceptual approach, the problem formulation, and the empirical attempts to validate spectral clustering on non-Euclidean manifolds.

**Claims And Evidence:**

No

**Claims Explanation:**

The theoretical claims made in the submission, particularly the proofs establishing weak consistency, contain fundamental mathematical flaws that remain unresolved in the revised manuscript. As rigorously detailed by the reviewers, the proof for Theorem 5.1 invalidly applies a 1-dimensional Euclidean identity (relating Wasserstein distance to the $L_1$ distance between CDFs) to random variables on a hyperbolic manifold, where such CDFs are not canonically defined. Furthermore, Theorem 5.2 retains an mathematically ill-defined statement regarding the covering number of the manifold with respect to an empirical function metric. Because the authors dismissed these valid critiques and failed to correct the underlying mathematical errors, the theoretical claims are not supported by accurate, convincing, or clear evidence.

**Resubmission Of Major Revision:**

The authors may consider submitting a major revision at a later time.